# An Efficient Bayesian Method for Estimating the Degree of the Skewness of X Chromosome Inactivation Based on the Mixture of General Pedigrees and Unrelated Females

**DOI:** 10.3390/biom13030543

**Published:** 2023-03-16

**Authors:** Yi-Fan Kong, Shi-Zhu Li, Kai-Wen Wang, Bin Zhu, Yu-Xin Yuan, Meng-Kai Li, Ji-Yuan Zhou

**Affiliations:** 1Department of Biostatistics, State Key Laboratory of Organ Failure Research, Ministry of Education, and Guangdong Provincial Key Laboratory of Tropical Disease Research, School of Public Health, Southern Medical University, Guangzhou 510515, China; 2Guangdong-Hong Kong-Macao Joint Laboratory for Contaminants Exposure and Health, Guangzhou 510006, China

**Keywords:** skewed X chromosome inactivation, Bayesian method, mixture of pedigrees and unrelated females, eigenvalue decomposition, Cholesky decomposition, Minnesota Center for Twin and Family Research data

## Abstract

Skewed X chromosome inactivation (XCI-S) has been reported to be associated with some X-linked diseases. Several methods have been proposed to estimate the degree of XCI-S (denoted as γ) for quantitative and qualitative traits based on unrelated females. However, there is no method available for estimating γ based on general pedigrees. Therefore, in this paper, we propose a Bayesian method to obtain the point estimate and the credible interval of γ based on the mixture of general pedigrees and unrelated females (called mixed data for brevity), which is also suitable for only general pedigrees. We consider the truncated normal prior and the uniform prior for γ. Further, we apply the eigenvalue decomposition and Cholesky decomposition to our proposed methods to accelerate the computation speed. We conduct extensive simulation studies to compare the performances of our proposed methods and two existing Bayesian methods which are only applicable to unrelated females. The simulation results show that the incorporation of general pedigrees can improve the efficiency of the point estimation and the precision and the accuracy of the interval estimation of γ. Finally, we apply the proposed methods to the Minnesota Center for Twin and Family Research data for their practical use.

## 1. Introduction

X chromosome inactivation (XCI) is an important epigenetic phenomenon, which was described by Lyon [1] for the first time. In mammals, females have two X chromosomes, whereas males have only one X chromosome. During the early development of embryos in females, one of the two X chromosomes becomes a Barr body and remains inactivated in subsequent somatic cells to ensure the balance of transcriptional dosages on the X chromosome between females and males [2]. In general, the process of XCI is random. Specifically, in females, approximately 50% of the cells have the paternal allele at an X-chromosomal locus inactivated, and the remaining approximately 50% of the cells keep the maternal allele inactivated, which is called random XCI (XCI-R) [3]. However, there are still two other patterns of XCI: the escape from XCI (XCI-E) and the skewed XCI (XCI-S) [3]. XCI-E means that a female has a region of the X chromosome without inactivation, i.e., the alleles on both X chromosomes are kept active. In humans, approximately 15–30% of X-linked loci have been reported to undergo XCI-E [4,5]. As for XCI-S, more than 75% of the cells inactivate the same allele at an X-chromosomal locus in females [6,7,8]. In some extremely skewed cases, it is possible that more than 90% of the cells have the same allele being inactivated [9].

Some X-linked diseases have been reported to be associated with XCI-S, such as esophageal carcinoma, recurrent spontaneous abortion, and Klinefelter’s syndrome [10,11,12,13]. The degree of XCI-S can affect the severity of X-linked diseases in heterozygous females [14]. A larger proportion of the cells with the activated deleterious allele in heterozygous females will cause more severe expression of the related diseases, whereas a smaller proportion can protect the females from the diseases [6,7]. For example, the XCI-S towards mutant alleles on the F9 gene may cause moderately severe haemophilia B, whereas the XCI-S against the same mutant alleles may cause mild haemophilia B in heterozygous females [15]. Thus, the incorporation of the XCI-S information into association analysis may improve the test power of the X-chromosomal association tests [16]. In fact, some methods have been proposed to test for the association between X-chromosomal single nucleotide polymorphisms (SNPs) and traits, which consider the XCI patterns [17,18,19,20,21,22,23,24,25,26]. For unrelated data, Wang et al. [24] proposed a permutation-based maximum likelihood ratio association test for qualitative traits, which takes account of all the XCI patterns. More specifically, for XCI, three female genotypes (dd, Dd, and DD) are encoded as 0, γ, and 2, respectively, meanwhile two male genotypes (d and D) are encoded as 0 and 2, respectively, where d and D are the normal and deleterious alleles, respectively. γ∈[0,2] is the unknown genotypic value used to measure the degree of XCI-S. For XCI-E, three female genotypes are encoded as 0, 1, and 2, and two male genotypes are encoded as 0 and 1. For pedigree data, Ding et al. [21] put forward a Monte Carlo pedigree disequilibrium test for X-linked qualitative traits and Zhang et al. [25] constructed the orthogonal model and used the kinship matrix to represent the correlation between the individuals in pedigrees for X-linked quantitative traits. Both methods take XCI-R or XCI-E into account, however, they are not suitable for XCI-S. Furthermore, the method of Ding et al. [21] cannot directly incorporate covariates and the method of Zhang et al. [25] is time-consuming. On the other hand, there is an autosomal association test, named GEMMA, which can incorporate covariates and is computationally efficient for pedigree data [27]. Moreover, GEMMA can be easily extended to accommodate the XCI-R and XCI-E patterns.

Recently, some methods to measure the degree of XCI-S have become available. Based on family trios (parents and their affected daughter), Xu et al. [28] proposed a statistical index to measure γ for qualitative traits. Wang et al. [29] and Li et al. [30] used unrelated females to estimate γ and derive the corresponding confidence interval (CI) for qualitative and quantitative traits, respectively. In Wang et al. [29] and Li et al. [30], γ was expressed as the ratio of two regression coefficients, and the CI was obtained using Fieller’s method. However, these methods may yield unbounded CIs when the denominator in the ratio is close to zero. It should be noted that Wang et al. [31] put forward a penalized Fieller’s method which can obtain the bounded CI of a ratio by penalizing the denominator of the ratio away from zero. Therefore, Yu et al. [32] applied the penalized Fieller’s method to the estimation of the degree of XCI-S for unrelated females. However, the penalized Fieller’s method does not consider the constraint condition of γ∈[0,2], and just simply uses the interval [0, 2] to truncate the point estimate and the CI of γ, which may result in extreme point estimates (0 or 2), empty sets, non-information intervals (i.e., [0, 2]), and discontinuous intervals. Therefore, Yu et al. [32] considered the constraint condition γ∈[0,2] as the prior, and further proposed a Bayesian method for estimating the degree of XCI-S based on unrelated females. The Bayesian method can avoid the generation of extreme point estimates, empty sets, non-information intervals, and discontinuous intervals. However, the above-mentioned methods are all based on family trios or unrelated females and cannot accommodate general pedigrees. It should be noted that general pedigrees are increasingly popular because pedigree designs are naturally equipped to control for population stratification [33,34]. Therefore, it is necessary to suggest a method for estimating the degree of XCI-S based on general pedigrees or the mixture of general pedigrees and unrelated females.

In this paper, we propose a Bayesian method to estimate the degree of XCI-S based on the mixture of general pedigrees and unrelated females for both quantitative and qualitative traits, which is also suitable for only general pedigrees. We use the kinship matrix to represent the correlation between females in general pedigrees and construct the generalized linear mixed model. The prior of γ is set to be a truncated normal distribution and a uniform distribution. The posterior distribution of γ is drawn using a Hamiltonian Monte Carlo (HMC) sampling algorithm. We regard the mode of the sample from the posterior distribution as the point estimate of γ, and consider the corresponding highest posterior density interval (HPDI) as the credible interval of γ [35]. Because the posterior sampling process of the generalized linear mixed model is very computationally intensive [36], we additionally employ the eigenvalue decomposition (EVD) and Cholesky decomposition to accelerate the computation speed. Further, we conduct extensive simulation studies to compare the performances of our proposed methods and the existing Bayesian methods. Finally, we apply our proposed methods to Minnesota Center for Twin and Family Research (MCTFR) data for their practical use.

## 2. Materials and Methods

### 2.1. Notations

Consider an X-chromosomal locus with alleles d and D being the normal allele and the deleterious allele, respectively. Suppose that we have collected the X-linked traits (quantitative or qualitative), the genotypes at the locus of $N_{p}$ pedigrees (including $n_{p}$ individuals, males or females), and additional $n_{If}$ independent/unrelated females. Note that the individuals in the same pedigree are genetically correlated. Since XCI only exists in females, we only select $n_{pf}$ females in these pedigrees and additional $n_{If}$ unrelated females to build the model, and we assume that $n_{f}=n_{pf}+n_{If}$. Let $Y_{i}$ be the trait of the ith female and $G_{i}=\{dd,Dd,DD\}$ indicate the genotype of the ith female $\left(i=1,2,\ldots ,n_{pf},n_{pf+1},\ldots ,n_{f}\right)$. Then, $Y={\left(Y_{1},Y_{2},\ldots ,Y_{n_{pf}},Y_{n_{pf}+1},\ldots ,Y_{n_{f}}\right)}^{T}$ is the trait vector of all the females, and $G={\left(G_{1},G_{2},\ldots ,G_{n_{pf}},G_{n_{pf}+1},\ldots ,G_{n_{f}}\right)}^{T}$ is the genotype vector of all the females. According to Wang et al. [24], we encode the genotypes $G_{i}=\{dd,Dd,DD\}$ as the genotypic values $X_{i}=\{0,\gamma ,2\}$, where γ∈[0,2] represents the degree of XCI-S. As such, the genotypic value vector of all the females can be expressed as $X={\left(X_{1},X_{2},\ldots ,X_{n_{pf}},X_{n_{pf}+1},\ldots ,X_{n_{f}}\right)}^{T}$. Considering the correlations among $n_{pf}$ females selected from $N_{p}$ pedigrees, we utilize the kinship matrix to measure the correlations of this kind. To be specific, we first use both the males and the females in the pedigrees to construct an $n_{p}\times n_{p}$ kinship matrix ψ, which can be obtained using the algorithm of Lange [37] through the “kinship2” package of R software [38]. Then, we select the corresponding rows and columns of $n_{pf}$ females in matrix ψ and obtain the $n_{pf}\times n_{pf}$ matrix $\psi_{f}$ of these $n_{pf}$ females. As for $n_{If}$ unrelated females, the genetic relatedness matrix can be expressed as the $n_{If}\times n_{If}$ identity matrix $I_{n_{If}\times n_{If}}$. Finally, the genetic relatedness matrix φ of Y can be denoted as the following block matrix:$$\varphi =\left(\begin{array}{cc} 2\psi_{f} & 0\\ 0 & I_{n_{If}\times n_{If}} \end{array}\right)$$

We build the generalized linear mixed model to describe the association between $G_{i}$ and $Y_{i}$
(1)$$h\left(\mu_{i}\right)=\beta X_{i}+a^{T}Z_{i}+b_{i}$$
where β is the regression coefficient of $X_{i}$; $Z_{i}={\left(Z_{i1},Z_{i2},\ldots ,Z_{im}\right)}^{T}$ is the vector of m covariates of the ith female including 1 as the first element and $Z={\left(Z_{1},Z_{2},\ldots ,Z_{n_{pf}},Z_{n_{pf}+1},\ldots ,Z_{n_{f}}\right)}^{T}$ is an $n_{f}\times m$ covariate matrix; $a={\left(a_{1},a_{2},\ldots ,a_{m}\right)}^{T}$ is the m×1 vector of the regression coefficients of $Z_{i}$ with $a_{1}$ being the intercept; $b_{i}$ is a random effect, and the random variable $b={\left(b_{1},b_{2},\ldots ,b_{n_{pf}},b_{n_{pf}+1},\ldots ,b_{n_{f}}\right)}^{T}$ is generated by the multivariate normal distribution, i.e., $b\sim MVN(0,\sigma_{g}^{2}\varphi )$, where $\sigma_{g}^{2}$ is the variance of the polygenic effects; h(⋅) is the link function; and $\mu_{i}=E\left(Y_{i}|X_{i},Z_{i}\right)$ is the conditional mean of $Y_{i}$ given $X_{i}$ and $Z_{i}$.

To estimate γ, we decompose $X_{i}$ in Equation (1) into $X_{i}=\gamma X_{1i}+(2-\gamma )X_{2i}$ according to Wang et al. [29], where $X_{1i}$ and $X_{2i}$ are two indicator variables. $X_{1i}=I_{\left\{G_{i}=DdorDD\right\}}$ indicates whether the ith female contains at least one deleterious allele D, and $X_{2i}=I_{\left\{G_{i}=DD\right\}}$ denotes whether the ith female has two deleterious alleles. Then, we can rewrite Equation (1) as follows:(2)$$h(\mu_{i})=\beta \gamma X_{1i}+\beta (2-\gamma )X_{2i}+a^{T}Z_{i}+b_{i}$$

For quantitative traits, h(⋅) is the identity function and $Y_{i}$ has the residual error $\varepsilon_{i}$, so Equation (2) becomes a linear mixed model
(3)$$Y_{i}=\beta \gamma X_{1i}+\beta (2-\gamma )X_{2i}+a^{T}Z_{i}+b_{i}+\varepsilon_{i}$$
where $\varepsilon_{i}\sim N(0,\sigma_{e}^{2})$ and $\sigma_{e}^{2}$ is the variance of $\varepsilon_{i}$. For qualitative traits, h(⋅) is the logit function, and Equation (2) can be written as
(4)$$logit(\mu_{i})=\beta \gamma X_{1i}+\beta (2-\gamma )X_{2i}+a^{T}Z_{i}+b_{i}$$

### 2.2. Building Bayesian Models

For quantitative traits, $Y={\left(Y_{1},Y_{2},\ldots ,Y_{n_{pf}},Y_{n_{pf}+1},\ldots ,Y_{n_{f}}\right)}^{T}$ follows a multivariate normal distribution according to Equation (3), i.e.,
(5)$$Y\sim MVN(\beta \gamma X_{1}+\beta (2-\gamma )X_{2}+Za,\sigma_{g}^{2}\varphi +\sigma_{e}^{2}I_{n_{f}\times n_{f}})$$
where $X_{1}={\left(X_{11},X_{12},\ldots ,X_{1n_{pf}},X_{1(n_{pf}+1)},\ldots ,X_{1n_{f}}\right)}^{T}$ and $X_{2}={\left(X_{21},X_{22},\ldots ,X_{2n_{pf}},X_{2(n_{pf}+1)},\ldots ,X_{2n_{f}}\right)}^{T}$. The unknown parameters are $\theta_{1}={\left(\beta ,\gamma ,a^{T},\sigma_{g},\sigma_{e}\right)}^{T}$, and let $L_{1}$ be the likelihood function of Y based on expression (5). So, the posterior distribution of $\theta_{1}$ can be expressed as $f\left(\theta_{1}|X_{1},X_{2},Z,\varphi \right)=\frac{f\left(\theta_{1}\right)L_{1}}{\int f\left(\theta_{1}\right)L_{1}d\theta_{1}}$, where $f\left(\theta_{1}\right)$ is the joint prior of $\theta_{1}$.

As for qualitative traits, $Y_{i}$ follows a Bernoulli distribution based on Equation (4), i.e.,
(6)$$Y_{i}\sim B(p_{i})$$
where $p_{i}={logit}^{-1}(\eta_{i})$ and
(7)$$\eta_{i}=\beta \gamma X_{1i}+\beta \left(2-\gamma \right)X_{2i}+a^{T}Z_{i}+b_{i}$$

The unknown parameters are $\theta_{2}={\left(\beta ,{\gamma ,a}^{T},\sigma_{g}\right)}^{T}$, and let $L_{2}$ be the likelihood function of Y based on expression (6). The posterior distribution of $\theta_{2}$ can be expressed as $f\left(\theta_{2}|X_{1},X_{2},Z,\varphi \right)=\frac{f\left(\theta_{2}\right)L_{2}}{\int f\left(\theta_{2}\right)L_{2}d\theta_{2}}$, where $f\left(\theta_{2}\right)$ is the joint prior of $\theta_{2}$.

### 2.3. Eigenvalue Decomposition and Cholesky Decomposition for Accelerating Computation Speed

It should be noted that, due to the high-dimensional matrix φ, the Bayesian posterior sampling processes of $f\left(\theta_{1}|X_{1},X_{2},Z,\varphi \right)$ and $f\left(\theta_{2}|X_{1},X_{2},Z,\varphi \right)$ would be computationally intensive, especially when $n_{f}$ is large [36,39]. So, according to Runcie and Crawford [40] and Zhao et al. [36], we use the EVD and Cholesky decomposition to accelerate the sampling process for quantitative and qualitative traits, respectively. The transformed posterior distributions of $\theta_{1}$ and $\theta_{2}$ are denoted by $f^{*}\left(\theta_{1}|X_{1}^{*},X_{2}^{*},Z^{*},\Sigma \right)$ and $f^{*}\left(\theta_{2}|X_{1},X_{2},Z,C,h\right)$, respectively, where $X_{1}^{*}=QX_{1}$, $X_{2}^{*}=QX_{2}$ and $Z^{*}=QZ$**,** respectively, are the transformed $X_{1}$, $X_{2}$ and Z based on $\varphi =Q^{T}\Sigma Q$ by the EVD; C is a lower triangular matrix satisfying $\varphi =CC^{T}$ by Cholesky decomposition; and h follows $MVN(0,I_{n_{f}\times n_{f}})$ and satisfies $\sigma_{g}Ch\sim MVN(0,\sigma_{g}^{2}\varphi )$. The details refer to Appendix A. From Table 1, we find that using the EVD and Cholesky decomposition in the posterior sampling process can greatly reduce running time (the details can be seen in Section 3).

### 2.4. HMC Algorithm and Priors

Note that it is difficult to derive the closed forms of the posterior distributions $f^{*}\left(\theta_{1}|X_{1}^{*},X_{2}^{*},Z^{*},\Sigma \right)$ and $f^{*}\left(\theta_{2}|X_{1},X_{2},Z,C,h\right)$, so we use the HMC algorithm [35] to sample the parameters from the approximate posterior distributions, which can be efficiently implemented through the “cmdstanr” package in R software. We choose the HMC algorithm because it can improve the independence of the samples and has higher efficiency than the other Markov-Chain Monte Carlo methods [35].

According to Yu et al. [32], we set the priors of $\theta_{1}$ and $\theta_{2}$ as follows: For nuisance parameters β and a, we select non-informative priors to reduce their influence on the posterior distributions. Specifically, we assume that $\beta \sim N(0,10^{2})$ and $a\sim MVN(0,diag(10^{2},10^{2},\ldots ,10^{2}))$ [41] so that β and a can be sampled from the positive and negative values with equal probabilities. For the standard deviation $\sigma_{g}$ of polygenic effects, we choose the exponential distribution with mean being 1, i.e., $\sigma_{g}\sim exp(1)$ [35]. For $\theta_{1}$ based on quantitative traits, there is an extra parameter $\sigma_{e}$, and we also suppose that $\sigma_{e}\sim exp(1)$. For the parameter γ of interest, by considering the constraint condition of γ∈[0,2], we set two priors. One is a uniform distribution from 0 to 2, i.e., γ~U(0,2), which is a non-informative prior. The other is to assume that the more skewed values of γ have the lower probability and the probability of γ being 1 is the highest, which is consistent with the genetic background [3]. In this way, we set γ to obey a truncated normal distribution with both the parameters being fixed at 1 and the values ranging from 0 to 2. The probability density function of the prior of γ is
$$f\left(\gamma \right)=\left\{\begin{array}{cc} \frac{\phi \left(\gamma -1\right)}{\Phi \left(1\right)-\Phi \left(-1\right)}, & 0\leq \gamma \leq 2\\ 0, & otherwise \end{array}\right.$$
where $\phi \left(\cdot \right)$ is the probability density function of the standard normal distribution and $\Phi \left(\cdot \right)$ is its cumulative distribution function. We assume that the unknown parameters are unrelated to each other because the HMC algorithm does not dramatically suffer from the correlated parameters in the model [35]. Therefore, the prior distributions $f\left(\theta_{1}\right)$ and $f\left(\theta_{2}\right)$ can be calculated as the product of the priors of all the parameters. Moreover, $f\left(\theta_{1}\right)$ and $f\left(\theta_{2}\right)$ can also be flexibly set according to practical background.

After we obtain the posterior samples of $\theta_{1}$ and $\theta_{2}$ through the HMC algorithm, we calculate the mode of the samples as the point estimate of γ, and compute the HPDI of the samples as the credible interval of γ. We denote the Bayesian methods with the truncated normal distribution and the uniform distribution for the mixture of general pedigrees and additional unrelated females as BNM and BUM, and the corresponding point estimates yielded by these two methods as ${\hat{\gamma }}_{BNM}$ and ${\hat{\gamma }}_{BUM}$, respectively.

### 2.5. Situations When Considering General Pedigrees and Unrelated Females, Respectively

Notice that our proposed methods are also applicable to the situation with only general pedigrees and that with only unrelated females. For the situation with only general pedigrees, the genetic relatedness matrix φ degenerates to twice the kinship matrix of all the $n_{pf}$ females from the pedigrees, i.e., $2\psi_{f}$. We denote the Bayesian methods with the truncated normal distribution and the uniform distribution for general pedigrees as BNP and BUP, and the corresponding point estimates as ${\hat{\gamma }}_{BNP}$ and ${\hat{\gamma }}_{BUP}$, respectively. For the situation with only unrelated females, our proposed methods still work where the genetic relatedness matrix φ is reduced to be the identity matrix $I_{n_{If}\times n_{If}}$. However, compared with the existing BN and BU methods having the prior of γ being the truncated normal distribution and the uniform distribution, respectively [32], our proposed methods require additionally estimating the random effects $b_{i}$’s, which may reduce the estimation accuracy and be time-consuming. Therefore, in practice, for unrelated females, we recommend using the existing BN and BU methods. Furthermore, just like Yu et al. [32], the point estimates of γ based on the BN and BU methods are represented as ${\hat{\gamma }}_{BN}$ and ${\hat{\gamma }}_{BU}$, respectively.

### 2.6. Situation When There Are Missing Genotypes for Some Individuals from General Pedigrees

It should be noted that our proposed methods are also suitable for the situation where the genotypes of some individuals from some pedigrees are missing, by simply excluding the individuals with missing genotypes and deleting the corresponding rows and columns of these individuals from the genetic relatedness matrix φ.

### 2.7. Simulation Settings

To evaluate the performance of our proposed methods (BNM and BUM for the mixture of general pedigrees and additional unrelated females, and BNP and BUP for only general pedigrees) and compare them with the existing methods (BN and BU for only unrelated females) [32] when estimating the degree of the XCI-S, we conduct the following extensive simulation studies. When simulating general pedigrees, we consider three pedigree structures: (1) the nuclear family with 4 people, (2) the three-generation family with 10 people and (3) the four-generation family with 12 people, as shown in Figure 1. We fix the sex ratio at 1:1 in our simulation study. A total of 50 pedigrees under each pedigree structure are simulated, which leads to $N_{p}$ being 150, $n_{p}$ being 1300, and $n_{pf}$ being approximately 650. For a larger sample size, we simulate 200 pedigrees under each pedigree structure, and the corresponding $N_{p}$ is 600, $n_{p}$ is 5200, and $n_{pf}$ is approximately 2600. Because there are two X chromosomes in females and only one in males, we first generate the genotypes {dd,Dd,DD} of the female founders using probabilities $\left\{{\left(1-p_{f}\right)}^{2},2p_{f}(1-p_{f}),p_{f}^{2}\right\}$ and the genotypes {d,D} of the male founders using probabilities $\left\{(1-p_{m}),p_{m}\right\}$, where $p_{f}$ and $p_{m}$ are the frequencies of the deleterious allele D in females and males, respectively. We first set $p_{f}$ to be 0.3 and 0.1 and keep $p_{m}$ consistent with $p_{f}$. To simulate the situations with $p_{f}$ and $p_{m}$ being different, we further set $\left(p_{f},p_{m}\right)=\left(0.3,0.1\right)$ and $\left(0.1,0.3\right)$. Then, we simulate the genotypes of the nonfounders according to Mendelian inheritance. We consider a covariate K, which is generated from the standard normal distribution. Note that the estimation of the degree of XCI-S only needs the females. As such, let $K_{pi}$ be the value of K for the ith female ($i=1,2,\ldots ,n_{pf}$) and we only simulate the quantitative trait values of all the $n_{pf}$ females in the pedigrees, which are generated based on the following multivariate normal distribution:(8)$$Y_{p}\sim MVN(\beta_{0}+\beta X_{p}+\delta K_{p},2\sigma_{g}^{2}\psi_{f}+\sigma_{e}^{2}I_{n_{pf}\times n_{pf}})$$
where $Y_{p}$ is the vector of the quantitative trait values of these $n_{pf}$ females; $X_{p}$ is the vector of their genotypic values with the elements being 0, γ, or 2 respectively corresponding to genotypes {dd,Dd,DD}, where the value of γ represents the degree of XCI-S and is randomly sampled from U(0,2); $K_{p}={\left(K_{p1},K_{p2},\ldots ,K_{pn_{pf}}\right)}^{T}$; and $\psi_{f}$ is the kinship matrix of the $n_{pf}$ females and $I_{n_{pf}\times n_{pf}}$ is an $n_{pf}\times n_{pf}$ identity matrix. $\beta_{0}$ is the intercept and δ is the regression coefficient of the covariate K, which are both fixed at 0.5 [42]. According to Schifano et al. [43], we set $\sigma_{g}^{2}=\{1/3,1\}$ and $\sigma_{e}^{2}=1$, which means that the values of the polygenic heritability $h_{p}^{2}=\sigma_{g}^{2}/(\sigma_{g}^{2}+\sigma_{e}^{2})=\{0.25,0.50\}$. Furthermore, we set β=0.2 so that the heritability due to the causal SNP, $h_{c}^{2}=\beta^{2}p_{f}(1-p_{f})/(\sigma_{g}^{2}+\sigma_{e}^{2})$, remains less than 2% for the chosen values of $p_{f}$, $\sigma_{g}^{2}$, and $\sigma_{e}^{2}$ mentioned above. As for a qualitative trait, we generate the corresponding values using the threshold model [44]. Specifically, once the quantitative trait values in Equation (8) are generated, they are transformed to be affected if they are less than the threshold and otherwise to be unaffected. Here, we fix the prevalence of the disease under study at 0.3, and the threshold is then taken as the 30% quantile of the distribution of the quantitative trait. In addition, to consider the situation in which the genotypes of some individuals in the pedigrees are missing, the missing rate (MR) is set to be 0 and 0.4. MR=0 means that the genotypes of all the individuals in the pedigrees are collected and MR=0.4 indicates that the genotype of an individual is randomly missing with probability 0.4.

When simulating unrelated females, we directly generate their genotypes {dd,Dd,DD} using probabilities $\left\{{\left(1-p_{f}\right)}^{2},2p_{f}(1-p_{f}),p_{f}^{2}\right\}$. For comparing BNP and BUP for only general pedigrees with BN and BU for only unrelated females, respectively, we set the number of unrelated females ($n_{If}$) to be 650 and 2600, which is almost equal to the number of the females in 150 and 600 pedigrees mentioned above, and we fix the variance of the residual error in the unrelated females at $\sigma_{g}^{2}+\sigma_{e}^{2}$ [45], which is the same as the total variance of the quantitative trait value in the females from the general pedigrees. Other parameters and simulation settings are kept the same as those when simulating general pedigrees. Specifically, the quantitative trait values of the $n_{If}$ unrelated females are generated according to the following multivariate normal distribution:$$Y_{I}\sim MVN(\beta_{0}+\beta X_{I}+\delta K_{I},(\sigma_{g}^{2}+\sigma_{e}^{2})I_{n_{If}\times n_{If}})$$
where $Y_{I}$ is the vector of the quantitative trait values of the $n_{If}$ unrelated females; $X_{I}$ is the vector of their genotypic values with the elements being 0, γ, or 2 corresponding to genotypes {dd,Dd,DD}; $K_{I}=(K_{I1},K_{I2},\ldots ,K_{In_{If}})$ is the covariate vector, where $K_{Ii}$ is the value of the covariate K for the ith female ($i=1,2,\ldots ,n_{If}$); and $I_{n_{If}\times n_{If}}$ is an $n_{If}\times n_{If}$ identity matrix. As for a qualitative trait, just like simulating the general pedigrees, we also generate the corresponding values using the threshold model [44]. By combining the females in the general pedigrees and additional unrelated females, we can obtain the mixed data. We use the BNM and BUM methods, the BNP and BUP methods, and the existing BN and BU methods to obtain the point estimates and the HPDIs of γ based on the mixed data, only general pedigrees, and only unrelated females, respectively.

Ma et al. [23] claimed that the variance of the quantitative trait under study for heterozygous females (Dd) may be higher than those for homozygous females (dd and DD) due to the XCI and other factors (e.g., gene-gene interactions and gene mutation), and the increase ratio can be up to 20%. However, so far, in our model, we do not consider the heteroscedasticity of this kind because of the potential computation cost in Bayesian inference. To investigate whether our proposed methods are still robust in the presence of the heteroscedasticity, we additionally simulate the mixed data for quantitative traits with the heteroscedasticity. Specifically, we use $\sigma_{e0}^{2}$, $\sigma_{e1}^{2}$, and $\sigma_{e2}^{2}$ to represent the residual variance $\sigma_{e}^{2}$ in females with genotypes dd, Dd, and DD, respectively. The simulation settings for the mixed data are the same as those under the homoscedasticity, except that we assume $\left(\sigma_{e0}^{2},\sigma_{e1}^{2},\sigma_{e2}^{2}\right)=(1,1.2,1)$ here. Furthermore, for comparison, we utilize $\left(\sigma_{e0}^{2},\sigma_{e1}^{2},\sigma_{e2}^{2}\right)=(1,1,1)$ to represent that the variances across different genotypes are the same. We apply the BNM and BUM methods to the mixed data, and apply the BNP and BUP methods to only general pedigrees.

We conduct 500 replicates for each simulation setting. For each replicate, we set 4 chains for extracting the samples simultaneously. For each chain, we extract 3000 samples, and the first 1000 samples are used for warming up. Therefore, we finally obtain 8000 samples in each replicate. To ensure the convergence, the target acceptance rate is taken as 0.9. We assess the convergence of Markov chains by calculating the convergence diagnostic $\hat{R}$ [46]. Note that the $\hat{R}$’s of our proposed methods are all less than 1.05, which indicates good convergence and also means that drawing 8000 samples is enough. The above posterior sampling process is implemented using the “cmdstanr” package in R software (version 4.1.2, http://r-project.org, accessed on 2 February 2023). To evaluate the accuracy of the point estimates, we calculate their mean squared errors (MSEs). Here, $MSE=\sum_{w=1}^{500}{\left({\hat{\gamma }}_{w}-\gamma_{w}\right)}^{2}/500$, where $\gamma_{w}$ is the wth true value of γ, and ${\hat{\gamma }}_{w}$ is the estimate of $\gamma_{w}$ (w=1,2,…,500). We also draw scatter plots to visually display the six point estimates (${\hat{\gamma }}_{BNM}$, ${\hat{\gamma }}_{BUM}$, ${\hat{\gamma }}_{BNP}$, ${\hat{\gamma }}_{BUP}$, ${\hat{\gamma }}_{BN}$, and ${\hat{\gamma }}_{BU}$) against the true values of γ. To compare the performances of the interval estimation of all the six methods (BNM, BUM, BNP, BUP, BN, and BU), we calculate the coverage probability (CP) as well as the median, the mean, the interquartile range, and the standard deviation of the widths of the 95% HPDIs of γ (respectively denoted by $W_{median}$, $W_{mean}$, $W_{iqr}$, and $W_{sd}$). Moreover, we draw scatter plots of the interval widths of all the six methods against the true values of γ.

## 3. Results

### 3.1. Simulation Results under the Situations of Homoscedasticity and Allele Frequencies in Females and Males Being the Same

To assess the computation efficiency of our proposed methods based on the EVD and Cholesky decomposition, we considered the BNP method for only general pedigrees as an example. Here, $N_{p}$ was taken to be 150 and 600, $p_{f}=p_{m}=0.3$, $\sigma_{g}^{2}=1/3$, $\left(\sigma_{e0}^{2},\sigma_{e1}^{2},\sigma_{e2}^{2}\right)=(1,1,1)$ and MR=0 (i.e., there were no missing genotypes in all the pedigrees) for both quantitative and qualitative traits. The other parameters were fixed in the same way as in the “Simulation Settings” subsection. A total of 500 replicates were conducted for each simulation setting. There were two kinds of BNP methods that we wanted to compare: (1) the BNP method with the posterior sampling process based on the EVD (for quantitative traits) or Cholesky decomposition (for qualitative traits), and (2) the BNP method with the posterior sampling process based on the posterior distribution $f\left(\theta_{1}|X_{1},X_{2},Z,\varphi \right)$ (for quantitative traits) or $f\left(\theta_{2}|X_{1},X_{2},Z,\varphi \right)$ (for qualitative traits), which is called the original posterior sampling process in this paper. We computed the mean running time of the BNP method based on the EVD or Cholesky decomposition for all 500 replicates. However, it is important to note that the original posterior sampling process may take up a huge amount of time. Therefore, we only calculated the mean running time of the original posterior sampling process over the first 10 replicates. All the computations were performed on a Tsinghua Tongfang Z900 personal computer (Microsoft Windows 7 Enterprise (Service Pack 1), 4 GB of RAM and 3.60 GHz Intel(R) Core(TM) i7-4790 CPU). The results of the mean running time are given in Table 1. As shown in Table 1, the EVD and Cholesky decomposition can greatly speed up the Bayesian sampling process, especially when $N_{p}$ is 600.

The MSEs of the six point estimates (${\hat{\gamma }}_{BNM}$, ${\hat{\gamma }}_{BUM}$, ${\hat{\gamma }}_{BNP}$, ${\hat{\gamma }}_{BUP}$, ${\hat{\gamma }}_{BN}$, and ${\hat{\gamma }}_{BU}$) of γ under $p_{f}=p_{m}$ and $\left(\sigma_{e0}^{2},\sigma_{e1}^{2},\sigma_{e2}^{2}\right)=(1,1,1)$ are listed in Table 2. We found that the MSEs of ${\hat{\gamma }}_{BNM}$ and ${\hat{\gamma }}_{BUM}$ based on the mixed data are the smallest under all the simulated scenarios, which means that it is more efficient to estimate the degree of XCI-S by simultaneously using general pedigrees and additional unrelated females. The MSEs of ${\hat{\gamma }}_{BNP}$ and ${\hat{\gamma }}_{BUP}$ for only general pedigrees are slightly larger than those of ${\hat{\gamma }}_{BN}$ and ${\hat{\gamma }}_{BU}$ for only unrelated females in all the simulated situations. This probably demonstrates that general pedigrees provide less information for estimating the degree of XCI-S than unrelated females when the total number of the females in all the pedigrees and that of the unrelated females are the same. As for the two priors of γ, the point estimates (${\hat{\gamma }}_{BNM}$, ${\hat{\gamma }}_{BNP}$, and ${\hat{\gamma }}_{BN}$) with the truncated normal distribution have the MSEs similar to those (${\hat{\gamma }}_{BUM}$, ${\hat{\gamma }}_{BUP}$, and ${\hat{\gamma }}_{BU}$) with the uniform distribution, with ${\hat{\gamma }}_{BNM}$, ${\hat{\gamma }}_{BNP}$, and ${\hat{\gamma }}_{BN}$ performing slightly better than ${\hat{\gamma }}_{BUM}$, ${\hat{\gamma }}_{BUP}$, and ${\hat{\gamma }}_{BU}$, respectively. Furthermore, it can be observed from Table 2 that the MSEs of the six point estimates decrease when $N_{p}$ and $n_{If}$ increase, $p_{f}$ and $p_{m}$ (the frequency of the deleterious allele D) increase, and $\sigma_{g}^{2}$ (the variance of the polygenic effects) decreases. As expected, compared to MR=0 (i.e., there are no missing genotypes in all the pedigrees), the MSEs of the six point estimates increase when MR=0.4 (i.e., the genotypes of about 40% individuals in general pedigrees are missing). In addition, the six point estimates have smaller MSEs for quantitative traits than for qualitative traits.

Figure 2 and Figure 3 show the scatter plots of the six point estimates (${\hat{\gamma }}_{BNM}$, ${\hat{\gamma }}_{BUM}$, ${\hat{\gamma }}_{BNP}$, ${\hat{\gamma }}_{BUP}$, ${\hat{\gamma }}_{BN}$, and ${\hat{\gamma }}_{BU}$) against the true values of γ with $N_{p}=150$, $n_{If}=650$, $p_{f}=p_{m}=0.3$, $\sigma_{g}^{2}=1/3$, $\left(\sigma_{e0}^{2},\sigma_{e1}^{2},\sigma_{e2}^{2}\right)=(1,1,1)$, and MR={0,0.4} for quantitative and qualitative traits, respectively. Appendix A show the corresponding scatter plots under other simulation settings. The six rows of each figure represent the results of the six point estimates, and the two columns of each figure denote the corresponding results with MR=0 and 0.4, respectively (i.e., subplots (a), (c), (e), (g), (i) and (k) are the scatter plots of ${\hat{\gamma }}_{BNM}$, ${\hat{\gamma }}_{BUM}$, ${\hat{\gamma }}_{BNP}$, ${\hat{\gamma }}_{BUP}$, ${\hat{\gamma }}_{BN}$, and ${\hat{\gamma }}_{BU}$ with MR=0, respectively, whereas subplots (b), (d), (f), (h), (j) and (l) are the corresponding scatter plots with MR=0.4). The upper side and the right side of each subplot are the distribution of the true value of γ and that of the point estimate of γ, respectively. By comparing the six subplots in the same column of each figure, we found that ${\hat{\gamma }}_{BNM}$ and ${\hat{\gamma }}_{BUM}$ based on the mixed data are closer to the true value of γ than ${\hat{\gamma }}_{BNP}$, ${\hat{\gamma }}_{BUP}$, ${\hat{\gamma }}_{BN}$, and ${\hat{\gamma }}_{BU}$. Moreover, noting that the distribution of the true value of γ is U(0,2), it can be seen that the distributions of ${\hat{\gamma }}_{BNM}$ and ${\hat{\gamma }}_{BUM}$ are more uniform than those of the four other point estimates. These indicate that it is necessary to combine general pedigrees with unrelated females when estimating γ. The dispersion of ${\hat{\gamma }}_{BNM}$ is slightly smaller than that of ${\hat{\gamma }}_{BUM}$, and the dispersions of ${\hat{\gamma }}_{BN}$ and ${\hat{\gamma }}_{BU}$ are slightly less than those of ${\hat{\gamma }}_{BNP}$ and ${\hat{\gamma }}_{BUP}$, although differences of these kinds are not so obvious in most figures. By comparing the two subplots in the same row of each figure, it can be seen that the estimates with MR=0.4 (in the subplot of the second column) have larger dispersion than those with MR=0 (in the subplot of the first column), implying that the missing genotypes of some individuals in the collected pedigrees would increase the MSEs of the point estimates. Furthermore, comparing Figure 2 with Figure 3 (or comparing Appendix A with Appendix A, respectively) shows that the six point estimates have better performance for quantitative traits than for qualitative traits. In addition, from these figures, the trend of the six point estimates with respect to $N_{p}$, $n_{If}$, $p_{f}$, $p_{m}$, and $\sigma_{g}^{2}$ is consistent with that in Table 2. Finally, it is observed from these figures that most of the point estimates can be evenly distributed on both sides of the true value of γ, except for the situations with $N_{p}=150$, $n_{If}=650$, and $p_{f}=p_{m}=0.1$, where the six point estimates may underestimate γ (Appendix A). However, when $N_{p}=600$, $n_{If}=2600$, and $p_{f}=p_{m}=0.1$, we can obtain point estimates which are much more evenly distributed around the true value of γ (Appendix A). This suggests that when analyzing the SNPs with low frequencies of the deleterious allele, our proposed point estimates need large sample sizes.

Table 3 describes the CPs of the six interval estimation methods (BNM, BUM, BNP, BUP, BN, and BU) under $p_{f}=p_{m}$ and $\left(\sigma_{e0}^{2},\sigma_{e1}^{2},\sigma_{e2}^{2}\right)=(1,1,1)$. From Table 3, we can find that all six methods can control the CPs around 95% in all the simulated situations, which verifies their accuracy when estimating the degree of XCI-S. Table 4 and Appendix A display the medians and the means of the widths of the 95% HPDIs ($W_{median}$ and $W_{mean}$), respectively, obtained by the six methods under $p_{f}=p_{m}$ and $\left(\sigma_{e0}^{2},\sigma_{e1}^{2},\sigma_{e2}^{2}\right)=(1,1,1)$. From these tables, we can see that the BNM and BUM methods based on the mixed data have smaller $W_{median}$ and $W_{mean}$ than the other four methods (BNP, BUP, BN, and BU) under all the simulated scenarios, which indicates that simultaneously using general pedigrees and additional unrelated females can improve the precision of the interval estimation of the degree of XCI-S. The $W_{median}$ and $W_{mean}$ of the BNP and BUP methods for only general pedigrees are slightly larger than those of the BN and BU methods for only unrelated females, which is consistent with the findings based on the MSEs of their corresponding point estimates from Table 2. For two priors of γ, the interval estimation with the truncated normal prior (the BNM, BNP, and BN methods) and that with the uniform prior (the BUM, BUP, and BU methods) have a similar performance, whereas the BNM, BNP, and BN methods respectively obtain slightly smaller $W_{median}$ and $W_{mean}$ than the BUM, BUP, and BU methods. When $N_{p}$ and $n_{If}$ increase, $p_{f}$ and $p_{m}$ (the frequency of the deleterious allele D) increase, $\sigma_{g}^{2}$ (the variance of the polygenic effects) decreases, MR (the probability of the genotype of an individual in a pedigree being missing) changes from 0.4 to 0, or the trait changes from qualitative to quantitative, the $W_{median}$ and $W_{mean}$ of the six methods decrease.

Table 5 and Appendix A show the interquartile range and the standard deviation of the widths of the 95% HPDIs ($W_{iqr}$ and $W_{sd}$), respectively, of the six methods under $p_{f}=p_{m}$ and $\left(\sigma_{e0}^{2},\sigma_{e1}^{2},\sigma_{e2}^{2}\right)=(1,1,1)$. Figure 4 and Figure 5 display the scatter plots of the widths of the 95% HPDIs based on the six interval estimation methods (BNM, BUM, BNP, BUP, BN, and BU) against the true values of γ with $N_{p}=150$, $n_{If}=650$, $p_{f}=p_{m}=0.3$, $\sigma_{g}^{2}=1/3$, $\left(\sigma_{e0}^{2},\sigma_{e1}^{2},\sigma_{e2}^{2}\right)=(1,1,1)$, and MR={0,0.4} for quantitative and qualitative traits, respectively. Appendix A give the corresponding scatter plots under other simulation settings. The six rows of each figure represent the results of the six methods, and the two columns of each figure denote the corresponding results with MR=0 and 0.4, respectively. From Table 5, we find that when the prior of γ is fixed to be the truncated normal distribution, the BNM method generally obtains smaller $W_{iqr}$ than the BNP and BN methods under all the simulated scenarios except for the cases of $N_{p}=150$, $n_{If}=650$, and $p_{f}=p_{m}=0.1$ for quantitative traits and those of $N_{p}=150$ and $n_{If}=650$ for qualitative traits. Similarly, when the prior of γ is taken as the uniform distribution, the $W_{iqr}$ of the BUM method are less than those of the BUP and BU methods under all the simulated scenarios in general, except for the situations mentioned above. It can be seen in Appendix A that the BNM (BUM) method generally derives smaller $W_{sd}$ than the BNP and BN (BUP and BU) methods except for the cases with $p_{f}=p_{m}=0.1$ for both quantitative and qualitative traits and those with $N_{p}=150$, $n_{If}=650$, and $p_{f}=p_{m}=0.3$ for qualitative traits. This may be explained by the fact that the largest width of the 95% HPDIs of the six methods is 2, and when $N_{p}=150$ and $n_{If}=650$ or $p_{f}=p_{m}=0.1$, the widths of the intervals obtained by the BNP, BUP, BN, and BU methods are very close to 2 (as can be observed in Appendix A), which make the dispersion of the widths of the intervals of the BNP, BUP, BN, and BU methods smaller and cause smaller $W_{iqr}$ and $W_{sd}$ of the BNP, BUP, BN, and BU methods. It is important to note that the width of the 95% HPDI of γ does not follow the normal distribution under most of the simulated scenarios (Appendix A), so the trend of the results of the $W_{sd}$ is not exactly the same as that of the $W_{iqr}$. On the other hand, the $W_{iqr}$ and $W_{sd}$ of the BNP (BUP) method are larger than those of the BN (BU) method. In addition, the $W_{iqr}$ and $W_{sd}$ of the six methods decrease with higher $p_{f}$ and $p_{m}$ when $N_{p}=600$ and $n_{If}=2600$, and increase when $N_{p}=150$ and $n_{If}=650$ and other parameters are unchanged. As for two priors, the BNM, BNP, and BN methods obtain slightly smaller $W_{iqr}$ and $W_{sd}$ than the BUM, BUP, and BU methods. It is shown in some subplots of Figure 4 and Figure 5 and Appendix A that the scatter plots look like an inverted V shape. This indicates that shorter intervals are obtained when the true values of γ are close to 0 or 2, by noting γ∈[0,2]. On the other hand, in some figures (e.g., Appendix A), most of the widths of the intervals based on the BNP, BUP, BN, and BU methods are very close to 2, which leads to the smaller dispersion of the interval widths. Other findings are similar to those from Table 4 and Table 5, and Appendix A, and we do not discuss them here for brevity.

### 3.2. Simulation Results When Allele Frequencies in Females and Males Being Different 

Appendix A shows the MSEs of the point estimates ${\hat{\gamma }}_{BNM}$, ${\hat{\gamma }}_{BUM}$, ${\hat{\gamma }}_{BNP}$, and ${\hat{\gamma }}_{BUP}$ under $\left(p_{f},p_{m}\right)=\left(0.3,0.1\right)$ and $\left(0.1,0.3\right)$, and $\left(\sigma_{e0}^{2},\sigma_{e1}^{2},\sigma_{e2}^{2}\right)=(1,1,1)$. Appendix A give the CP, $W_{median}$, $W_{mean}$, $W_{iqr}$, and $W_{sd}$, respectively, of the BNM, BUM, BNP, and BUP methods under $\left(p_{f},p_{m}\right)=\left(0.3,0.1\right)$ and $\left(0.1,0.3\right)$, and $\left(\sigma_{e0}^{2},\sigma_{e1}^{2},\sigma_{e2}^{2}\right)=(1,1,1)$. It can be observed from Appendix A that when $\left(p_{f},p_{m}\right)=\left(0.3,0.1\right)$ and $\left(0.1,0.3\right)$, all four methods control the CPs around 95%. From Appendix A, the MSE, $W_{median}$, $W_{mean}$, $W_{iqr}$ and $W_{sd}$ of the four methods with $\left(p_{f},p_{m}\right)=\left(0.3,0.1\right)$ and $\left(0.1,0.3\right)$ are generally smaller than those with $\left(p_{f},p_{m}\right)=\left(0.1,0.1\right)$ and larger than those with $\left(p_{f},p_{m}\right)=\left(0.3,0.3\right)$ (compared with Table 2 and Table 4, Appendix A, Table 5 and Appendix A, respectively), implying that our proposed methods still work when there are differences in the frequencies of the deleterious alleles between females and males.

### 3.3. Simulation Results under Heteroscedasticity 

Appendix A displays the MSEs of the point estimates ${\hat{\gamma }}_{BNM}$, ${\hat{\gamma }}_{BUM}$, ${\hat{\gamma }}_{BNP}$, and ${\hat{\gamma }}_{BUP}$ under $p_{f}=p_{m}$, and $\left(\sigma_{e0}^{2},\sigma_{e1}^{2},\sigma_{e2}^{2}\right)=(1,1,1)$ and (1,1.2,1). Appendix A show the CP, $W_{median}$, $W_{mean}$, $W_{iqr}$, and $W_{sd}$, respectively, of the BNM, BUM, BNP, and BUP methods under $p_{f}=p_{m}$, and $\left(\sigma_{e0}^{2},\sigma_{e1}^{2},\sigma_{e2}^{2}\right)=(1,1,1)$ and (1,1.2,1). As shown in Appendix A, our four proposed methods all control the CPs around 95% well when heteroscedasticity exists (i.e., $\left(\sigma_{e0}^{2},\sigma_{e1}^{2},\sigma_{e2}^{2}\right)=(1,1.2,1)$). From Appendix A, we can find that the MSE, $W_{median}$,$W_{mean}$, $W_{iqr}$, and $W_{sd}$ of our proposed methods under heteroscedasticity are similar to the corresponding results under homoscedasticity (i.e., $\left(\sigma_{e0}^{2},\sigma_{e1}^{2},\sigma_{e2}^{2}\right)=(1,1,1)$) for all simulated situations, which indicates that our proposed methods are still robust when heteroscedasticity is present.

### 3.4. Application to MCTFR Data

The MCTFR Genome-Wide Association Study of Behavioral Disinhibition is a family-based study of substance abuse and related psychopathology [47]. The dataset can be downloaded from the database of Genotypes and Phenotypes with the accession number phs000620.v1.p1 (https://www.ncbi.nlm.nih.gov/projects/gap/cgi-bin/study.cgi?study_id=phs000620.v1.p1, accessed on 2 February 2023). This dataset contains 2183 families, 7377 participants (3831 females and 3546 males), and 527,829 SNPs. There are five quantitative traits in the dataset: the nicotine composite score, the alcohol consumption composite score, the alcohol dependence composite score (DEP), the illicit drug composite score, and the behavioral disinhibition composite score [48]. Because we only use females for measuring the degree of XCI-S, 3831 females and 12,354 SNPs on the X chromosome were selected. We filtered the data using the following quality control criteria: (1) excluding SNPs with a missing rate > 10%, (2) removing SNPs with a minor allele frequency < 5%, and (3) excluding individuals with a genotype missing rate > 10%. After quality control, 850 families, 3195 females (including 1959 females from 850 families and additional 1236 unrelated females), and 11,344 SNPs were kept to conduct the subsequent analyses.

It is important to note that estimating γ requires the SNPs on the X chromosome to be associated with the traits under study. Therefore, borrowing the idea of the GEMMA method for association analysis on autosomes based on only general pedigrees [27], we propose an improved linear mixed model to test for association on the X chromosome based on the mixed data. We made the following two main modifications: Firstly, we set the relatedness matrix as the block matrix φ in the Materials and Methods section so that the proposed linear mixed model is applicable to the mixture of general pedigrees and additional unrelated females. Secondly, the parameter γ is generally unknown. To consider the XCI, referring to Wang et al. [24], we utilized the grid search method and γ was taken to be {0, 0.5, 1, 1.5, 2} in the increments of 0.5. We used the improved linear mixed model to calculate the *p*-value for each value of γ, and then combined these five *p*-values using Cauchy’s method [49] to obtain the final test statistic. We conducted some simulation studies and found that the proposed improved linear mixed model can control the type I error rate well (the details can be seen in Appendix A). It should be noted that the five quantitative traits in the MCTFR dataset do not follow normal distributions. Therefore, we transformed the traits using the rank-based inverse normal transformation [50] before carrying out association analysis. Furthermore, we incorporated two covariates, age and year of birth, into the improved linear mixed model. The significance level of the association tests was set to be $0.05/11344=4.41\times 10^{-6}$ based on the Bonferroni correction.

The proposed linear mixed model identified three SNPs, rs10522027, rs12860832, and rs12849233, which are associated with the DEP trait at the $4.41\times 10^{-6}$ level. The positions, alleles, minor allele frequencies, corresponding traits, *p*-values, and genes which the three SNPs belong to are presented in Table 6. SNP rs10522027 is found within the gene transmembrane protein 47 (TMEM47), which may be associated with the chemoresistance of breast cancer cells and hepatocellular carcinoma [51]. SNPs rs12860832 and rs12849233 are found in the gene PAS domain containing repressor 1 (PASD1), which might serve as a new target for the prognosis and the future treatment of glioma [52]. Furthermore, we calculated the point estimates (${\hat{\gamma }}_{BNM}$, ${\hat{\gamma }}_{BUM}$, ${\hat{\gamma }}_{BNP}$, ${\hat{\gamma }}_{BUP}$, ${\hat{\gamma }}_{BN}$, and ${\hat{\gamma }}_{BU}$) and the 95% HPDIs of γ based on the proposed BNM, BUM, BNP, and BUP methods and the existing BN and BU methods for these three SNPs, where the BNM and BUM methods use the mixed data (850 families and an additional 1236 unrelated females), the BNP and BUP methods utilize only 850 families with 1959 females, and the BN and BU methods are applied to only the additional 1236 unrelated females. The point estimates and the corresponding 95% HPDIs of γ obtained by the six methods for these SNPs are listed in Table 7. It is shown that the six point estimates of γ for the three SNPs are not far away from one, and the corresponding 95% HPDIs all contain one, which means that the XCI patterns of the three SNPs are the XCI-R or the XCI-E. In addition, we can observe the advantage of the BNM and BUM methods because they generally obtain smaller credible intervals than the other four methods, which is consistent with our simulation results. However, the BNP and BUP methods can give shorter HPDIs than the BN and BU methods, which does not coincide with our simulation results. This could be because the number of females in the 850 families is 1959, which is much larger than the number of additional unrelated females (1236).

## 4. Discussion

In this paper, we consider a generalized linear mixed model and propose two Bayesian methods (BNM and BUM) to estimate the degree of XCI-S (i.e., γ) based on the mixture of general pedigrees and additional unrelated females for both quantitative and qualitative traits, where the BNM method uses the prior of the truncated normal distribution and the BUM method utilizes the prior of the uniform distribution, which both make full use of the constraint condition of γ∈[0,2]. When only general pedigrees were available, the BNM and BUM methods were reduced to the BNP and BUP methods, respectively. We do not propose the corresponding Fieller’s method and the Penalized Fieller’s method to estimate the degree of XCI-S based on general pedigrees in this paper, as it has been confirmed that the performance of the above two methods is worse than Bayesian methods for only unrelated females [32]. It is important to note that that the closed form of the posterior distribution of γ is not easily derived, so we applied the HMC algorithm to conduct the posterior sampling process, calculated the mode of the resulting samples as the point estimate of γ, and regarded the HPDI of γ as the credible interval of γ. However, the posterior sampling process based on general pedigrees is very computationally intensive, especially when the dimension of the relatedness matrix (i.e., φ in this paper) is over 1000 [36]. As such, we used the EVD and Cholesky decomposition of φ to speed up the posterior sampling process for quantitative and qualitative traits, respectively. On the other hand, we also considered the median and the percentile interval (the 2.5th and 97.5th percentiles) of the posterior samples as the point estimate and the credible interval of γ, respectively. However, they performed less well than the mode and the HPDI (data not shown for brevity), and then we selected the latter instead.

The simulation results demonstrate that the EVD and Cholesky decomposition can greatly speed up the posterior sampling process, which is important to allow our proposed methods to accommodate large sample sizes, and may be referenced by other Bayesian researchers. The simulation results under $p_{f}=p_{m}$ and $\left(\sigma_{e0}^{2},\sigma_{e1}^{2},\sigma_{e2}^{2}\right)=(1,1,1)$ also show that the BNM and BUM methods have similar performances and are advantageous over the other four methods, which indicates that it is necessary to simultaneously analyze general pedigrees and additional unrelated females when estimating the degree of XCI-S in practice. More specifically, for the point estimation, the MSEs of ${\hat{\gamma }}_{BNM}$ and ${\hat{\gamma }}_{BUM}$ are close to each other and are smaller than those of the four other point estimates. The MSE of ${\hat{\gamma }}_{BNM}$ is the smallest in all the simulated situations. The MSEs of the existing point estimates ${\hat{\gamma }}_{BN}$ and ${\hat{\gamma }}_{BU}$ for unrelated females are slightly smaller than those of ${\hat{\gamma }}_{BNP}$ and ${\hat{\gamma }}_{BUP}$ for general pedigrees when the number of females is fixed. This suggests that general pedigrees provide less information for estimating the degree of XCI-S than unrelated females when the total number of the females in all the pedigrees and that of the unrelated females are the same. For the interval estimation, all six methods (BNM, BUM, BNP, BUP, BN, and BU) control the CPs around 95%. The BNM and BUM methods perform similarly to each other and both obtain much smaller credible intervals ($W_{median}$ and $W_{mean}$) than the other four methods under all the simulated scenarios. The BNP and BUP methods perform slightly worse than the BN and BU methods when the number of females is fixed, which is consistent with the findings based on the point estimation (${\hat{\gamma }}_{BNP}$, ${\hat{\gamma }}_{BUP}$, ${\hat{\gamma }}_{BN}$, and ${\hat{\gamma }}_{BU}$). For two priors of γ, the performances of the BNM, BNP, and BN methods with the truncated normal prior are slightly better than those of the BUM, BUP, and BU methods with the uniform prior, whereas differences of these kinds are not so obvious in our proposed methods, suggesting that our proposed methods are not as sensitive to the choice of priors. Furthermore, our proposed methods perform better when $N_{p}$ and $n_{If}$ increase, $p_{f}$ and $p_{m}$ (the frequency of the deleterious allele D) increase, $\sigma_{g}^{2}$ (the variance of the polygenic effects) decreases, or the trait changes from qualitative to quantitative. When there are missing genotypes for some individuals in pedigrees, the SLINK software based on the peeling algorithm [53] could be used to impute these missing genotypes. However, to make the test statistics in hypothesis testing robust, or the parameter estimation accurate and precise, one may repeatedly impute the missing genotypes using the SLINK software (e.g., 50 imputations), which is very time-consuming for our proposed Bayesian methods. On the other hand, it is easy to combine 50 resulting point estimates of γ by taking the mean, median, or mode of them as the final point estimate; however, there appears to be an issue with the process of combining the 50 resulting credible intervals. Therefore, when the genotypes of some individuals in the collected pedigrees were missing, we did not impute these missing genotypes. Instead, we chose to delete the individuals with missing genotypes directly. In fact, the simulation results show that, even when the genotypes of approximately 40% of individuals in general pedigrees are missing, our proposed methods can still control the CPs well, indicating that our proposed methods are robust when there are missing genotypes in the data. The simulation results also show that our proposed methods still work when the frequency of the deleterious allele in females and that in males are different (i.e., $\left(p_{f},p_{m}\right)=\left(0.3,0.1\right)$ and $\left(0.1,0.3\right)$). Furthermore, when heteroscedasticity exists (i.e., $\left(\sigma_{e0}^{2},\sigma_{e1}^{2},\sigma_{e2}^{2}\right)=(1,1.2,1)$), our proposed methods remain robust.

The proposed methods have the following issues to be discussed: Firstly, it is well known that the prior distributions of unknown parameters are important in Bayesian inference, and the choice of them may affect the results. In this paper, we consider two priors for γ, a non-informative prior U(0,2) which has little effect on the posterior distribution of γ, and a truncated normal distribution $N\left(1,1\right)\in [0,2]$ based on the genetic background of XCI. We also take account of non-informative priors for regression coefficients and weak priors for variances. In practical applications, researchers can choose appropriate priors according to their research background. Secondly, the Bayesian method adopts the HMC algorithm for the posterior sampling process, which is not greatly influenced by the correlations among unknown parameters. Therefore, for computational efficiency, we assume that all unknown parameters are unrelated. However, Bayesian methods should have better performance if the correlation between parameters is considered. Thirdly, the HPDIs that contain the number one can only indicate that the SNP undergoes the XCI-R or XCI-E pattern. The process of further distinguishing the XCI-R and XCI-E patterns is a potential problem to be solved. Fourthly, Ma et al. [23] claimed that the variance of the quantitative trait under study for heterozygous females may be higher than that for homozygous females in some cases. For computational efficiency, we assumed that the variances of quantitative traits for different genotypes in females are the same in our proposed methods. 

To address the issues mentioned above, we will consider the following improvements in the future: Firstly, we will take into account non-informative priors for variances, such as non-informative Gamma prior or inverse-Gamma prior [41], to improve our proposed methods. Secondly, we will use the Gibbs sampling algorithm [54] to conduct the Bayesian posterior sampling process when the parameters are correlated. Thirdly, the information from the XCI-E can be estimated using the difference of transcriptional dosage on the X chromosome between male hemizygotes and female homozygotes. Therefore, we will incorporate the information from males into our model to further distinguish the XCI-E from the XCI-R. Fourthly, although we have completed some simulation studies showing that our proposed methods are still robust in the presence of heteroscedasticity (Appendix A), we will extend our proposed methods to manage the situation of heteroscedasticity to further improve the precision and the accuracy of estimating the degree of XCI-S in the future. Finally, besides the GEMMA [27], we understand that the REGENIE method for autosomal SNPs [55] could take into account population stratification. Therefore, we will extend it to test for the association between X chromosomal SNPs and traits based on the mixed data in the future.

## 5. Conclusions

In summary, we propose a Bayesian method with two priors (the truncated normal prior and the uniform prior) to estimate the degree of XCI-S based on the mixture of general pedigrees and additional unrelated females, which are denoted by the BNM and BUM methods, respectively. We also develop the corresponding Bayesian method, which is suitable for only general pedigrees, denoted by the BNP and BUP methods. We conducted an extensive simulation study to compare the performance of our four proposed methods with the two existing BN and BU methods. The simulation results show that the BNM method obtains the smallest MSE, the shortest width of the HPDIs, and the most stable CPs, which indicates that it is more efficient in estimating the degree of XCI-S by simultaneously using general pedigrees and additional unrelated females. Finally, we applied the proposed methods to the MCTFR data, and found that three associated SNPs (rs10522027, rs12860832, and rs12849233) undergo the XCI-R or XCI-E pattern.

## Figures and Tables

**Figure 1 biomolecules-13-00543-f001:**
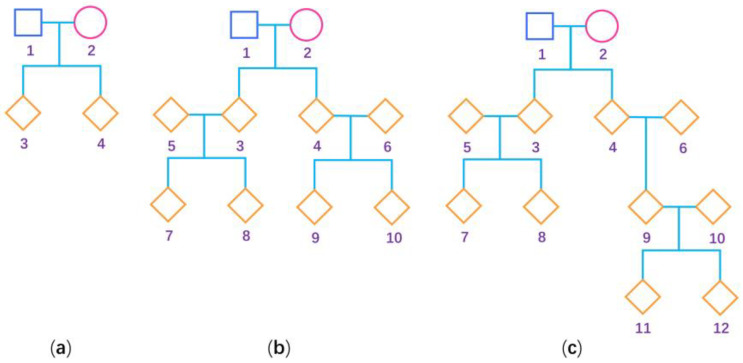
Pedigree structure used for the simulation studies. The squares are males, the circles are females and the rhombus could be any gender. The numbers are used to encode the family members. (**a**) Nuclear family; (**b**) three-generation family; and (**c**) four-generation family.

**Figure 2 biomolecules-13-00543-f002:**
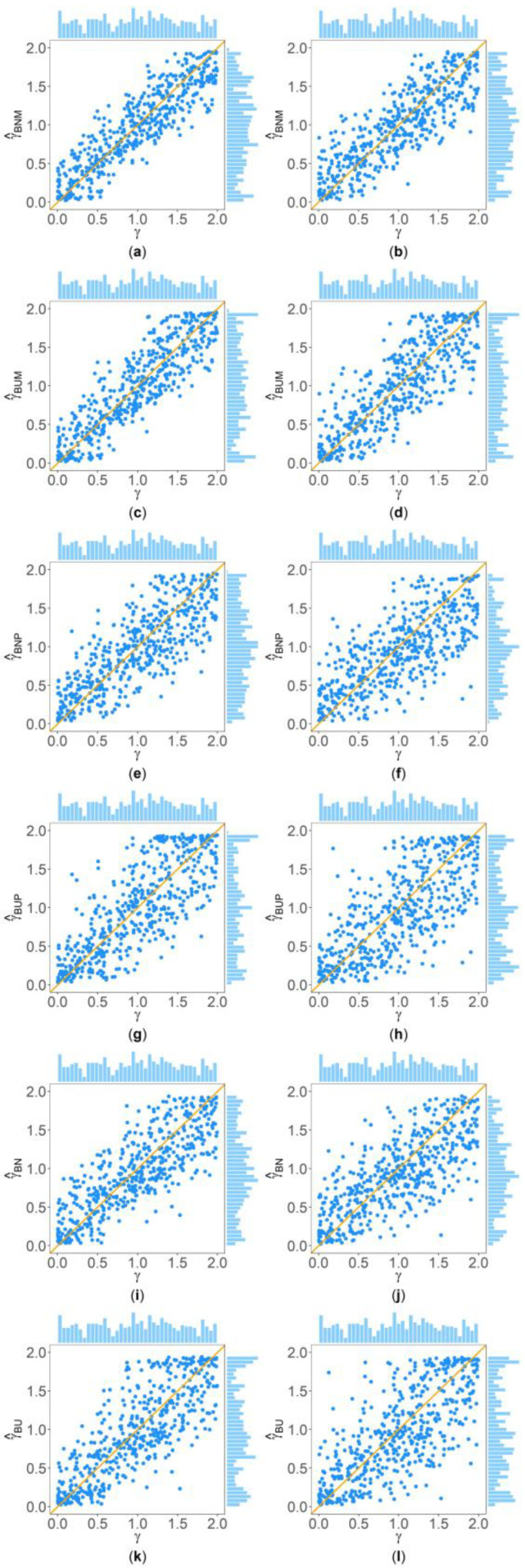
Scatter plots of six point estimates of γ against true values of γ with $N_{p}=150$, $n_{If}=650$, $p_{f}=p_{m}=0.3$, $\sigma_{g}^{2}=1/3$, $\left(\sigma_{e0}^{2},\sigma_{e1}^{2},\sigma_{e2}^{2}\right)=(1,1,1)$, and MR={0,0.4} for quantitative trait. The upper side and the right side of each subplot are the distribution of the true value of γ and that of the point estimate of γ, respectively. (**a**) ${\hat{\gamma }}_{BNM}$ with MR=0; (**b**) ${\hat{\gamma }}_{BNM}$ with MR=0.4; (**c**) ${\hat{\gamma }}_{BUM}$ with MR=0; (**d**) ${\hat{\gamma }}_{BUM}$ with MR=0.4; (**e**) ${\hat{\gamma }}_{BNP}$ with MR=0; (**f**) ${\hat{\gamma }}_{BNP}$ with MR=0.4; (**g**) ${\hat{\gamma }}_{BUP}$ with MR=0; (**h**) ${\hat{\gamma }}_{BUP}$ with MR=0.4; (**i**) ${\hat{\gamma }}_{BN}$ with MR=0; (**j**) ${\hat{\gamma }}_{BN}$ with MR=0.4; (**k**) ${\hat{\gamma }}_{BU}$ with MR=0; and (**l**) ${\hat{\gamma }}_{BU}$ with MR=0.4.

**Figure 3 biomolecules-13-00543-f003:**
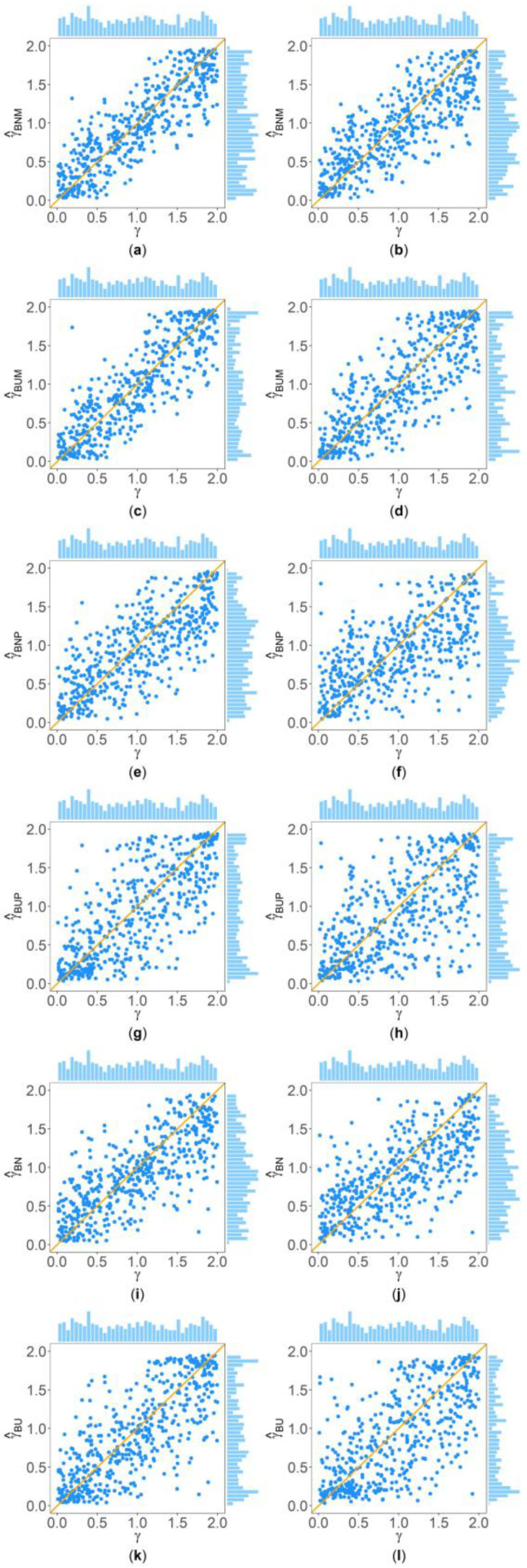
Scatter plots of six point estimates of γ against true values of γ with $N_{p}=150$, $n_{If}=650$, $p_{f}=p_{m}=0.3$, $\sigma_{g}^{2}=1/3$, $\left(\sigma_{e0}^{2},\sigma_{e1}^{2},\sigma_{e2}^{2}\right)=(1,1,1)$, and MR={0,0.4} for qualitative trait. The upper side and the right side of each subplot are the distribution of the true value of γ and that of the point estimate of γ, respectively. (**a**) ${\hat{\gamma }}_{BNM}$ with MR=0; (**b**) ${\hat{\gamma }}_{BNM}$ with MR=0.4; (**c**) ${\hat{\gamma }}_{BUM}$ with MR=0; (**d**) ${\hat{\gamma }}_{BUM}$ with MR=0.4; (**e**) ${\hat{\gamma }}_{BNP}$ with MR=0; (**f**) ${\hat{\gamma }}_{BNP}$ with MR=0.4; (**g**) ${\hat{\gamma }}_{BUP}$ with MR=0; (**h**) ${\hat{\gamma }}_{BUP}$ with MR=0.4; (**i**) ${\hat{\gamma }}_{BN}$ with MR=0; (**j**) ${\hat{\gamma }}_{BN}$ with MR=0.4; (**k**) ${\hat{\gamma }}_{BU}$ with MR=0; and (**l**) ${\hat{\gamma }}_{BU}$ with MR=0.4.

**Figure 4 biomolecules-13-00543-f004:**
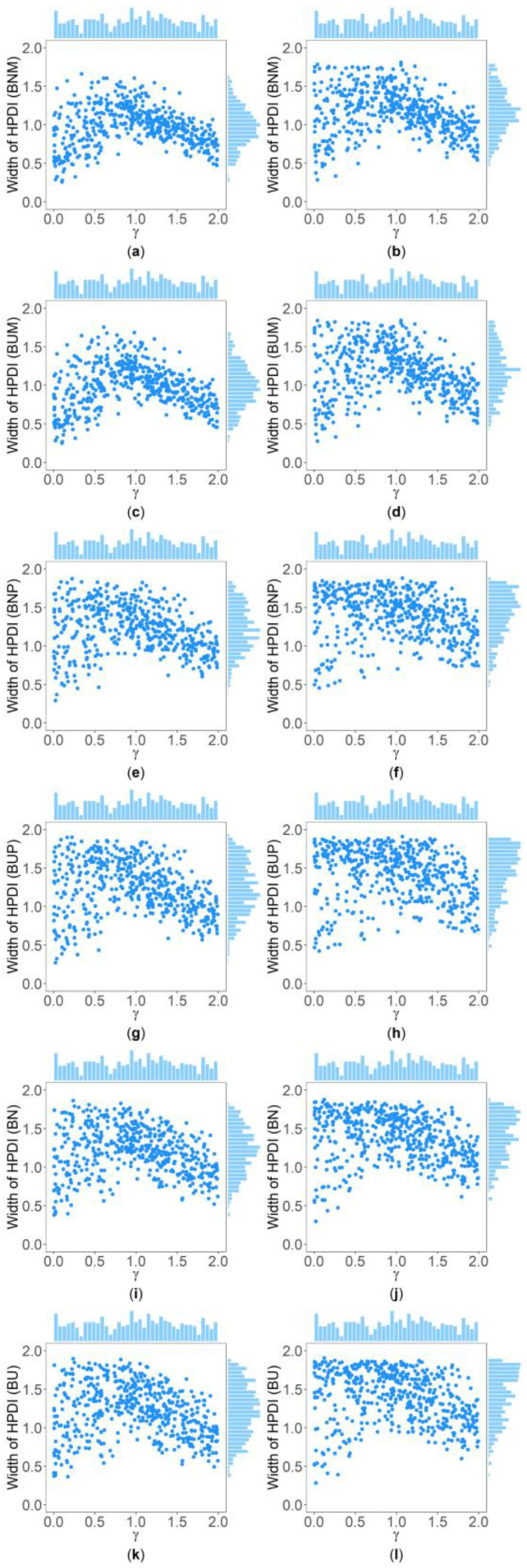
Scatter plots of widths of HPDIs based on six methods against true values of γ with $N_{p}=150$, $n_{If}=650$, $p_{f}=p_{m}=0.3$, $\sigma_{g}^{2}=1/3$, $\left(\sigma_{e0}^{2},\sigma_{e1}^{2},\sigma_{e2}^{2}\right)=(1,1,1)$, and MR={0,0.4} for quantitative trait. The upper side and the right side of each subplot are the distribution of the true value of γ and that of the width of the HPDI of γ, respectively. (**a**) BNM with MR=0; (**b**) BNM with MR=0.4; (**c**) BUM with MR=0; (**d**) BUM with MR=0.4; (**e**) BNP with MR=0; (**f**) BNP with MR=0.4; (**g**) BUP with MR=0; (**h**) BUP with MR=0.4; (**i**) BN with MR=0; (**j**) BN with MR=0.4; (**k**) BU with MR=0; and (**l**) BU with MR=0.4.

**Figure 5 biomolecules-13-00543-f005:**
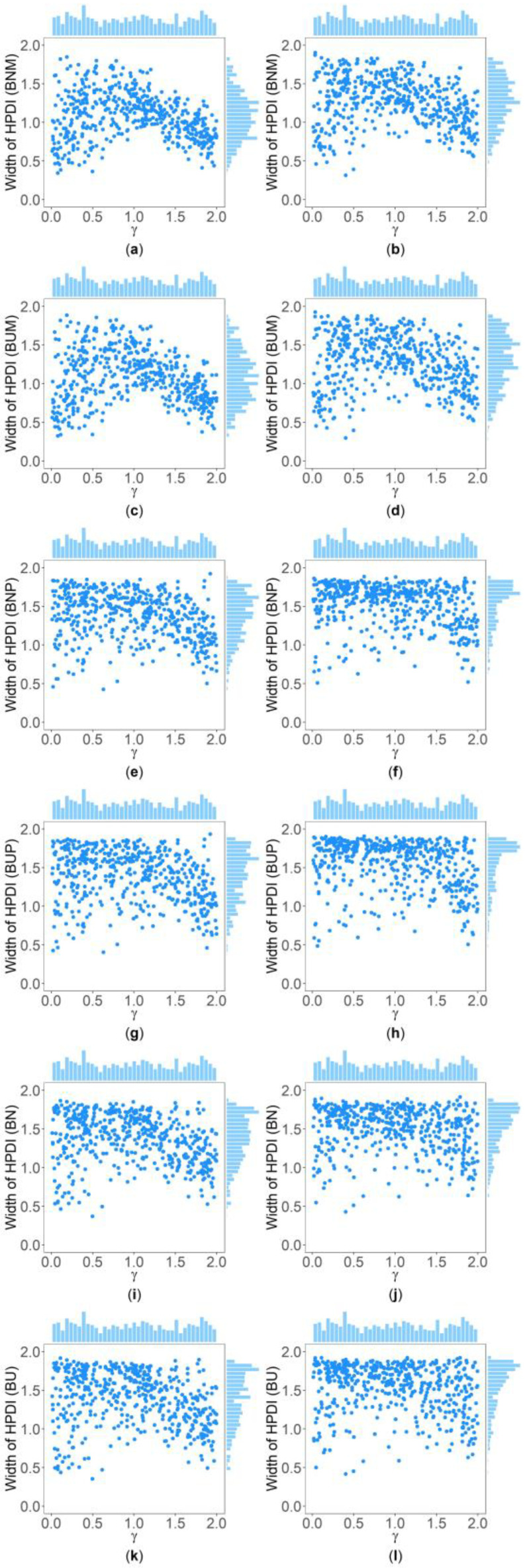
Scatter plots of widths of HPDIs based on six methods against true values of γ with $N_{p}=150$, $n_{If}=650$, $p_{f}=p_{m}=0.3$, $\sigma_{g}^{2}=1/3$, $\left(\sigma_{e0}^{2},\sigma_{e1}^{2},\sigma_{e2}^{2}\right)=(1,1,1)$, and MR={0,0.4} for qualitative trait. The upper side and the right side of each subplot are the distribution of the true value of γ and that of the width of the HPDI of γ, respectively. (**a**) BNM with MR=0; (**b**) BNM with MR=0.4; (**c**) BUM with MR=0; (**d**) BUM with MR=0.4; (**e**) BNP with MR=0; (**f**) BNP with MR=0.4; (**g**) BUP with MR=0; (**h**) BUP with MR=0.4; (**i**) BN with MR=0; (**j**) BN with MR=0.4; (**k**) BU with MR=0; and (**l**) BU with MR=0.4.

**Table 1 biomolecules-13-00543-t001:** Mean running time of the BNP method with a posterior sampling process based on EVD or Cholesky decomposition and an original posterior sampling process for general pedigrees.

$N_{p}$	Quantitative Trait		Qualitative Trait	
	EVD ^a^	Original ^b^	Cholesky ^a^	Original ^b^
150	10 s	1.8 h	1.3 min	7.2 h
600	40 s	7 days	47 min	30 days

^a^ The mean running time is based on 500 replicates; ^b^ the mean running time is based on 10 replicates.

**Table 2 biomolecules-13-00543-t002:** Mean squared errors (MSEs) of point estimates ${\hat{\gamma }}_{BNM}$, ${\hat{\gamma }}_{BUM}$, ${\hat{\gamma }}_{BNP}$, ${\hat{\gamma }}_{BUP}$, ${\hat{\gamma }}_{BN}$, and ${\hat{\gamma }}_{BU}$ under $p_{f}=p_{m}$ and $\left(\sigma_{e0}^{2},\sigma_{e1}^{2},\sigma_{e2}^{2}\right)=(1,1,1)$ among 500 replicates for mixed data, only general pedigrees, and only unrelated females, respectively.

Trait	$\left(N_{p},n_{If}\right)$	$p_{f}$	$\sigma_{g}^{2}$	MR	Mixed Data		Pedigrees		Unrelated Females	
					${\hat{\gamma }}_{BNM}$	${\hat{\gamma }}_{BUM}$	${\hat{\gamma }}_{BNP}$	${\hat{\gamma }}_{BUP}$	${\hat{\gamma }}_{BN}$	${\hat{\gamma }}_{BU}$
Quantitative	(150, 650)	0.3	1/3	0	0.0643	0.0707	0.1167	0.1342	0.1123	0.1258
		0.3	1/3	0.4	0.0943	0.1032	0.1564	0.1757	0.1532	0.1704
		0.3	1	0	0.0889	0.0966	0.1528	0.1646	0.1391	0.1556
		0.3	1	0.4	0.1323	0.1473	0.2086	0.2409	0.2017	0.2354
		0.1	1/3	0	0.1850	0.1968	0.2959	0.3247	0.2284	0.2499
		0.1	1/3	0.4	0.2455	0.2670	0.3781	0.4325	0.3304	0.3706
		0.1	1	0	0.2010	0.2192	0.3399	0.3792	0.3260	0.3595
		0.1	1	0.4	0.2754	0.3064	0.4224	0.4900	0.4046	0.4709
	(600, 2600)	0.3	1/3	0	0.0229	0.0236	0.0407	0.0421	0.0377	0.0383
		0.3	1/3	0.4	0.0359	0.0377	0.0570	0.0606	0.0558	0.0596
		0.3	1	0	0.0256	0.0260	0.0543	0.0576	0.0540	0.0571
		0.3	1	0.4	0.0416	0.0445	0.0805	0.0879	0.0764	0.0813
		0.1	1/3	0	0.0786	0.0846	0.1210	0.1300	0.1169	0.1209
		0.1	1/3	0.4	0.1147	0.1205	0.1689	0.1750	0.1593	0.1684
		0.1	1	0	0.0962	0.1039	0.1650	0.1773	0.1512	0.1660
		0.1	1	0.4	0.1353	0.1415	0.2080	0.2252	0.1910	0.2056
Qualitative	(150, 650)	0.3	1/3	0	0.0862	0.0926	0.1551	0.1774	0.1499	0.1621
		0.3	1/3	0.4	0.1243	0.1298	0.2181	0.2593	0.1922	0.2197
		0.3	1	0	0.1175	0.1249	0.2112	0.2381	0.1923	0.2138
		0.3	1	0.4	0.1588	0.1863	0.2768	0.3405	0.2459	0.2793
		0.1	1/3	0	0.2654	0.2815	0.3911	0.4343	0.3822	0.4107
		0.1	1/3	0.4	0.4051	0.4250	0.5403	0.6122	0.5383	0.6103
		0.1	1	0	0.2910	0.3316	0.4342	0.5159	0.4322	0.4921
		0.1	1	0.4	0.4430	0.5020	0.6024	0.6967	0.5978	0.6838
	(600, 2600)	0.3	1/3	0	0.0344	0.0347	0.0551	0.0599	0.0540	0.0562
		0.3	1/3	0.4	0.0564	0.0581	0.0847	0.0921	0.0811	0.0885
		0.3	1	0	0.0441	0.0459	0.0816	0.0872	0.0667	0.0700
		0.3	1	0.4	0.0727	0.0743	0.1154	0.1261	0.1045	0.1091
		0.1	1/3	0	0.1092	0.1214	0.2405	0.2546	0.1549	0.1615
		0.1	1/3	0.4	0.1577	0.1665	0.3585	0.3722	0.2225	0.2273
		0.1	1	0	0.1429	0.1496	0.2427	0.2570	0.1718	0.1832
		0.1	1	0.4	0.1783	0.2017	0.3614	0.3857	0.2472	0.2614

**Table 3 biomolecules-13-00543-t003:** Coverage probabilities (CPs, in %) of the BNM, BUM, BNP, BUP, BN, and BU methods under $p_{f}=p_{m}$ and $\left(\sigma_{e0}^{2},\sigma_{e1}^{2},\sigma_{e2}^{2}\right)=(1,1,1)$ among 500 replicates for mixed data, only general pedigrees, and only unrelated females, respectively ^a^.

Trait	$\left(N_{p},n_{If}\right)$	$p_{f}$	$\sigma_{g}^{2}$	MR	Mixed Data		Pedigrees		Unrelated Females	
					BNM	BUM	BNP	BUP	BN	BU
Quantitative	(150, 650)	0.3	1/3	0	94.6	96.2	95.4	96.2	95.2	94.8
		0.3	1/3	0.4	95.0	95.6	95.2	95.6	95.4	96.0
		0.3	1	0	95.0	95.0	93.8	94.6	95.2	95.2
		0.3	1	0.4	94.4	95.2	94.2	95.0	94.0	94.4
		0.1	1/3	0	93.8	93.6	95.8	94.4	95.0	95.2
		0.1	1/3	0.4	93.4	95.4	95.0	94.2	94.4	95.4
		0.1	1	0	96.2	94.6	94.0	95.0	94.2	94.8
		0.1	1	0.4	94.4	95.8	94.2	95.4	93.8	94.2
	(600, 2600)	0.3	1/3	0	94.2	94.2	94.0	94.4	94.8	95.0
		0.3	1/3	0.4	94.2	94.4	95.6	95.4	95.6	95.4
		0.3	1	0	95.0	96.4	95.6	95.8	95.2	95.2
		0.3	1	0.4	95.8	95.8	94.6	94.6	95.4	94.8
		0.1	1/3	0	94.2	94.2	95.0	94.6	94.4	95.4
		0.1	1/3	0.4	94.0	95.0	94.6	95.2	95.2	96.2
		0.1	1	0	95.6	95.6	95.6	96.0	93.6	95.2
		0.1	1	0.4	95.0	95.4	94.8	95.6	95.6	94.6
Qualitative	(150, 650)	0.3	1/3	0	95.8	94.8	95.0	94.8	94.2	95.2
		0.3	1/3	0.4	94.8	95.0	94.2	95.0	94.6	95.2
		0.3	1	0	95.2	95.6	95.4	94.8	95.0	95.6
		0.3	1	0.4	94.8	96.4	94.2	95.4	94.8	93.8
		0.1	1/3	0	95.4	94.8	95.2	94.8	95.4	94.6
		0.1	1/3	0.4	95.2	95.0	94.8	95.2	94.8	95.6
		0.1	1	0	93.6	94.6	95.0	95.8	95.4	94.8
		0.1	1	0.4	94.8	95.2	94.6	94.8	95.0	95.4
	(600, 2600)	0.3	1/3	0	95.4	94.8	95.0	94.8	96.2	95.6
		0.3	1/3	0.4	93.8	95.2	94.2	95.4	95.0	94.2
		0.3	1	0	94.2	94.2	95.0	94.8	94.2	95.4
		0.3	1	0.4	95.6	95.2	95.0	95.6	95.8	94.4
		0.1	1/3	0	93.6	94.6	96.0	94.4	94.6	95.8
		0.1	1/3	0.4	95.0	95.0	94.4	95.6	94.0	95.0
		0.1	1	0	94.2	93.8	95.2	93.4	95.2	95.0
		0.1	1	0.4	95.6	95.2	94.4	95.4	95.0	95.8

^a^ The empirical CP should be between 93.05% and 96.95% ($0.95\pm 2\times \sqrt{\frac{0.95\times 0.05}{500}}$) with 95% probability.

**Table 4 biomolecules-13-00543-t004:** $W_{median}$s of the BNM, BUM, BNP, BUP, BN, and BU methods under $p_{f}=p_{m}$ and $\left(\sigma_{e0}^{2},\sigma_{e1}^{2},\sigma_{e2}^{2}\right)=(1,1,1)$ among 500 replicates for mixed data, only general pedigrees, and unrelated females, respectively.

Trait	$\left(N_{p},n_{If}\right)$	$p_{f}$	$\sigma_{g}^{2}$	MR	Mixed Data		Pedigrees		Unrelated Females	
					BNM	BUM	BNP	BUP	BN	BU
Quantitative	(150, 650)	0.3	1/3	0	0.9770	0.9815	1.2152	1.2336	1.2103	1.2249
		0.3	1/3	0.4	1.1601	1.1801	1.4467	1.4781	1.3937	1.4373
		0.3	1	0	1.0627	1.0636	1.3667	1.3966	1.3405	1.3653
		0.3	1	0.4	1.2572	1.2627	1.5525	1.6017	1.5260	1.5821
		0.1	1/3	0	1.4258	1.4452	1.5863	1.6305	1.5729	1.6297
		0.1	1/3	0.4	1.5502	1.5935	1.6720	1.7201	1.6584	1.7103
		0.1	1	0	1.5453	1.5940	1.6713	1.7166	1.6637	1.7112
		0.1	1	0.4	1.6493	1.6999	1.7109	1.7633	1.7106	1.7629
	(600, 2600)	0.3	1/3	0	0.5350	0.5378	0.7332	0.7324	0.7292	0.7278
		0.3	1/3	0.4	0.6658	0.6642	0.8894	0.8840	0.8639	0.8650
		0.3	1	0	0.6216	0.6272	0.8754	0.8861	0.8388	0.8433
		0.3	1	0.4	0.7755	0.7868	1.0389	1.0515	1.0265	1.0282
		0.1	1/3	0	1.0073	1.0217	1.2340	1.2505	1.2201	1.2450
		0.1	1/3	0.4	1.1494	1.1790	1.3633	1.3885	1.3538	1.3814
		0.1	1	0	1.1208	1.1298	1.3468	1.3654	1.3392	1.3562
		0.1	1	0.4	1.3067	1.3313	1.4664	1.5054	1.4634	1.5038
Qualitative	(150, 650)	0.3	1/3	0	1.0719	1.0821	1.4111	1.4353	1.3991	1.4332
		0.3	1/3	0.4	1.2857	1.3103	1.6186	1.6573	1.5767	1.6258
		0.3	1	0	1.2224	1.2394	1.5703	1.6178	1.5559	1.6176
		0.3	1	0.4	1.4206	1.4531	1.6704	1.7288	1.6644	1.7193
		0.1	1/3	0	1.5167	1.5485	1.6562	1.7093	1.6551	1.6972
		0.1	1/3	0.4	1.5914	1.6449	1.7038	1.7543	1.6725	1.7236
		0.1	1	0	1.6121	1.6757	1.7135	1.7649	1.6907	1.7450
		0.1	1	0.4	1.6751	1.7284	1.7203	1.7690	1.7160	1.7642
	(600, 2600)	0.3	1/3	0	0.6888	0.6920	0.8648	0.8684	0.8369	0.8448
		0.3	1/3	0.4	0.8207	0.8274	1.0530	1.0555	1.0117	1.0057
		0.3	1	0	0.7781	0.7777	1.0235	1.0199	0.9527	0.9559
		0.3	1	0.4	0.9714	0.9739	1.1988	1.2089	1.1522	1.1660
		0.1	1/3	0	1.1414	1.1550	1.3164	1.3447	1.3027	1.3380
		0.1	1/3	0.4	1.3028	1.3184	1.4463	1.4831	1.4182	1.4421
		0.1	1	0	1.2614	1.2848	1.4063	1.4336	1.4022	1.4294
		0.1	1	0.4	1.4044	1.4328	1.5227	1.5681	1.5185	1.5555

**Table 5 biomolecules-13-00543-t005:** $W_{iqr}$s of the BNM, BUM, BNP, BUP, BN, and BU methods under $p_{f}=p_{m}$ and $\left(\sigma_{e0}^{2},\sigma_{e1}^{2},\sigma_{e2}^{2}\right)=(1,1,1)$ among 500 replicates for mixed data, only general pedigrees, and only unrelated females, respectively.

Trait	$\left(N_{p},n_{If}\right)$	$p_{f}$	$\sigma_{g}^{2}$	MR	Mixed Data		Pedigrees		Unrelated Females	
					BNM	BUM	BNP	BUP	BN	BU
Quantitative	(150, 650)	0.3	1/3	0	0.3333	0.3700	0.4620	0.5268	0.4530	0.5122
		0.3	1/3	0.4	0.4106	0.4680	0.4760	0.5466	0.4435	0.4987
		0.3	1	0	0.4218	0.4755	0.4739	0.5418	0.4291	0.5261
		0.3	1	0.4	0.4351	0.5176	0.4162	0.4673	0.4069	0.4314
		0.1	1/3	0	0.4058	0.4620	0.3309	0.3603	0.3193	0.3392
		0.1	1/3	0.4	0.3389	0.3812	0.2603	0.2690	0.2267	0.2243
		0.1	1	0	0.3345	0.3817	0.2612	0.2806	0.2586	0.2585
		0.1	1	0.4	0.2467	0.2585	0.1618	0.1658	0.1590	0.1488
	(600, 2600)	0.3	1/3	0	0.1590	0.1629	0.2391	0.2655	0.2272	0.2582
		0.3	1/3	0.4	0.2184	0.2344	0.3295	0.3803	0.2823	0.3190
		0.3	1	0	0.1946	0.2087	0.3241	0.3795	0.2968	0.3297
		0.3	1	0.4	0.2733	0.3080	0.3892	0.4434	0.3643	0.4125
		0.1	1/3	0	0.3785	0.4380	0.3878	0.4345	0.3790	0.4319
		0.1	1/3	0.4	0.4386	0.4744	0.3998	0.4649	0.3880	0.4338
		0.1	1	0	0.3744	0.4297	0.4101	0.4489	0.3670	0.4449
		0.1	1	0.4	0.3653	0.3961	0.3640	0.4160	0.3524	0.4152
Qualitative	(150, 650)	0.3	1/3	0	0.4290	0.4922	0.4841	0.5399	0.4561	0.5148
		0.3	1/3	0.4	0.4542	0.5314	0.3988	0.4512	0.3902	0.4319
		0.3	1	0	0.4539	0.5204	0.4009	0.4466	0.3970	0.4309
		0.3	1	0.4	0.4424	0.5062	0.2980	0.3132	0.2893	0.3005
		0.1	1/3	0	0.3811	0.4325	0.2866	0.3086	0.2694	0.2752
		0.1	1/3	0.4	0.3118	0.3636	0.2822	0.3342	0.2245	0.2540
		0.1	1	0	0.3195	0.3387	0.2239	0.2082	0.1856	0.1867
		0.1	1	0.4	0.2468	0.2913	0.1973	0.2209	0.1932	0.2027
	(600, 2600)	0.3	1/3	0	0.1886	0.2109	0.2929	0.3232	0.2715	0.3036
		0.3	1/3	0.4	0.2580	0.2884	0.3947	0.4478	0.3536	0.4152
		0.3	1	0	0.2680	0.2937	0.3801	0.4009	0.3617	0.3981
		0.3	1	0.4	0.3576	0.4061	0.4404	0.5107	0.4060	0.4630
		0.1	1/3	0	0.3826	0.4368	0.3421	0.4035	0.3350	0.3903
		0.1	1/3	0.4	0.3333	0.3700	0.4620	0.5268	0.4530	0.5122
		0.1	1	0	0.4106	0.4680	0.4760	0.5466	0.4435	0.4987
		0.1	1	0.4	0.4218	0.4755	0.4739	0.5418	0.4291	0.5261

**Table 6 biomolecules-13-00543-t006:** SNPs detected in association analysis for the MCTFR data.

SNP	Position	Alleles	MAF ^a^	Trait	*p*-Value	Gene
rs10522027	34630163	G > A	0.141	DEP	${3.64\times 10}^{-7}$	TMEM47
rs12860832	151643064	G > A	0.263	DEP	${2.00\times 10}^{-6}$	PASD1
rs12849233	151645704	C > A	0.329	DEP	${1.26\times 10}^{-6}$	PASD1

^a^ MAF represents the minor allele frequency.

**Table 7 biomolecules-13-00543-t007:** Application of the six methods to SNPs detected in association analysis for the MCTFR data.

SNP	Point Estimate						95% HPDI					
	${\hat{\gamma }}_{BNM}$	${\hat{\gamma }}_{BUM}$	${\hat{\gamma }}_{BNP}$	${\hat{\gamma }}_{BUP}$	${\hat{\gamma }}_{BN}$	${\hat{\gamma }}_{BU}$	BNM	BUM	BNP	BUP	BN	BU
rs10522027	0.6922	0.6895	0.6394	0.6494	0.7238	0.7429	(0.2451, 1.3518)	(0.2316, 1.4420)	(0.0156, 1.5816)	(0.0063, 1.5567)	(0.1791, 1.6615)	(0.1870, 1.6384)
rs12860832	0.8371	0.8288	0.9422	0.9448	0.7281	0.7200	(0.3266, 1.4935)	(0.3942, 1.5788)	(0.1878, 1.6258)	(0.2077, 1.6698)	(0.0945, 1.6294)	(0.1214, 1.6503)
rs12849233	0.7633	0.7426	0.8843	0.8736	0.6906	0.6968	(0.2236, 1.2934)	(0.2133, 1.3054)	(0.1054, 1.5392)	(0.1361, 1.5964)	(0.0211, 1.5229)	(0.0764, 1.5490)

## Data Availability

The R package BEMXCIS for the BNM, BUM, BNP and BUP methods is freely available at https://github.com/Yi-FanKong/BEMXCIS (accessed on 2 February 2023), which is implemented by R software (version 4.1.2). The MCTFR data used for this study can be found on the database of Genotypes and Phenotypes (dbGaP) with the accession number phs000620.v1.p1 and the dbGaP request number 86747-7 (https://www.ncbi.nlm.nih.gov/projects/gap/cgi-bin/study.cgi?study_id=phs000620.v1.p1, accessed on 2 February 2023).

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
