# Peer review of "An Efficient Bayesian Method for Estimating the Degree of the Skewness of X Chromosome Inactivation Based on the Mixture of General Pedigrees and Unrelated Females"

_biomolecules, 2023, doi:10.3390/biom13030543_

Round 1
Reviewer 1 Report
In this paper, the authors proposed a Bayesian method for estimating the degree of the skewness of X Chromosome Inactivation. The statistical calculations are solid and the simulation studies and real data analysis is very convincing. In general this is a very interesting paper. I only have a few minor comments:
1. The method part is too long. Please consider to move some technical details to the online supplementary.
2. section 2.3, the reason of using EVD is to speed up the MCMC process when the parameters are correlated. I'm wondering have you consider the Gibbs sampling algorithm as an alternative? Please discuss.
3. Page 5 line 225, "weak prior". better use "non-informative prior".
4. Page 4 line 229, exp(1) is kind of informative prior, have you considered non-informative Gamma prior or inverse-Gamma prior? Please discuss.
Reviewer 2 Report
In this manuscript, the authors proposed a Bayesian method to obtain the point estimate and the credible interval of the degree of XCI-S (denoted as ?) for the mixture of general pedigrees and unrelated females. The computational speed was further improved by applying the eigenvalue decomposition and Cholesky decomposition. Both simulation study and real data analysis showed the advantages of the proposed methods. In general, I think that the manuscript is well-written and the research topic is interesting. My comments are detailed below:
1. The authors claimed that when the genotypes of some individuals from some pedigrees are missing, they simply excluded the individuals with missing genotypes and deleted the corresponding rows and columns of these individuals from the genetic relatedness matrix φ to estimate ?. In fact, the SLINK software based on the peeling algorithm (Weeks, D.E.; Ott, J.; Lathrop, G.M. SLINK: a general simulation program for linkage analysis. Am. J. Hum. Genet. 1990, 47, A204) could be used to impute the missing genotypes. The authors could use this software to impute missing genotypes which may improve the performance of the estimation of ?.
2. The authors respectively regarded the mode and the highest posterior density interval (HPDI) of the samples from the posterior distribution of gamma as the point estimates and the credible intervals of gamma. Have the authors considered using the median and the percentile interval (the 2.5th ~ 97.5th percentiles) of the samples? How about their performance?
3. Lines 367 and 368 on Page 8: both f(θ_1│X_1, X_2, Z, φ) and f(θ_2│X_1, X_2, Z, φ) are the posterior distributions of the corresponding parameters. So, the words “the likelihood function” should be changed to “posterior distribution”.
4. It may be more appropriate to move the “Simulation Settings” subsection from the “Results” section to the “Materials and Methods” section.
Reviewer 3 Report
An Efficient Bayesian Method for Estimating the Degree of the Skewness of X Chromosome Inactivation Based on the Mixture of General Pedigrees and Unrelated Females
In this work the authors present a new method for the estimation of the degree of XCI skewness, that could be used to improve GWAS results on chrX. The idea and the novelty of the work resides in the ability to apply this new method to a generic cohort of samples, with feasible computational time.
The paper is generally well written and worth of publication.
I have some major points for witch I’d like the authors clarification:
1) I feel that the term “general pedigree” is a little confusing for the reader. Since the authors want to point out the integration between pedigree data from trios/families and unrelated samples, why they are not referring to the “mixed data” as “cohort based pedigree” or something similar as opposite to family based pedigree and unrelated samples cohort?
2) Why extend the model of the GEMMA software and not of something like REGENIE (https://doi.org/10.1038/s41588-021-00870-7) or SAIGE (https://doi.org/10.1038/s41588-018-0184-y ), that should already be able to take into account the population stratification in the study cohort? This way the authors could have only devised the integration of gamma in the model without taking into account the shape of the kinship matrix. Is it only due to the fact that the GEMMA framework is easily extendable ?
3) In the “Application to MCTFR data” paragraph, the authors state that they did “some simulation studies” to support the good control for type I error on their proposed modified model, but there is no mention of it in the supplementary materials. Can they clarify this statement in the text?
Below are also some minor observations that I feel the authors could take into account to improve the overall quality of the text:
Abstract:
line 17: would say “several methods have been proposed”
line 18: would remove the “currently” that seems out of context
Introduction:
line 36: would use “described” instead of “Proposed”
line 42: “another about 50%” should be changed
line 87: since it seems that ref [31] is not strictly related to the estimation of gamma, I would not use the expression “On the other hand” because it could be confusing for the reader.
M&M:
line 232: would remove the “for it” statement since it is not needed
Results:
the “Simulation settings” paragraph should go under the M&M section
lines 281-283: is not clear what the numbers in the round brackets are
lines 288-289: it would be better to have all the results here, instead of putting them in the discussion.
line 341: would use only “simultaneously” and remove the redundant “parallelly”
Simulation results:
line 384: would remove the repeated “from table 2”
line 394: would change “while …. perform” with “with …. performing” for better clarity
line 418: “moreover” instead of “On the other hand”
line 434: “smaller” instead of less
line 435: “is consistent” instead of “are consistent”
In general the description of the comparison between figures in lines 430-435 is a little too convoluted to read, moreover, the same conclusion can be drawn from the data presented in table 2 and supplementary tables. A more compact description could improve the paper readability.
Lines 490-496: the description of the pictures here is a repetition of lines 410-415 (though the axis represent different things, the concept is the same and this detailed description could be kept in the figure caption)
line 507: change “it” with “this”
line 511: should change “less” with “small” or “smaller”
Application to MCTFR data:
lines 555-556: would use “excluding SNPs with missing rate ...” instead of “excluding those SNPs with the missing rate…” and “removing SNPs with minor allele frequency...” instead of “deleting those SNPs with the minor allele frequency…” and “excluding individuals with genotype missing rate …” instead of “excluding those individuals with the genotype missing rate...”
Line 596: change the “that” with “since” and “shorter” with “smaller”
line 599: change “should” with “could”
Discussion:
line 631: would change “much less” with “smaller”
line 635: would remove “probably”
line 639: would change “have the performance” to “have performances”
line 640: change “shorter” with “smaller”
lines 659-672: these could be in the result section, leaving only the final comment for the discussion.
Line 673: the expression “on the other hand”, at this point, seems out of context
line 681-693: as for lines 659-672 this information should be given in the result section
line 698-709: the authors are basically reporting again the results of their method applied to the MCTFR cohort, It would be better to avoid such repetition in the discussion.
Lines 710-731: the form of the text should be improved for better clarity, maybe adding a separate paragraph for the “future work” statements.
